# The Noisy Laplacian: a Threshold Phenomenon for Non-Linear Dimension Reduction

Alex Kokot [1]   Octavian-Vlad Murad [2]   Marina Meila [1]

## Abstract

In this paper, we clarify the effect of noise on common spectrally motivated algorithms such as Diffusion Maps (DM) for dimension reduction. Empirically, these methods are much more robust to noise than current work suggests. Specifically, existing consistency results require that either the noise amplitude or dimensionality must vary with the sample size $n$. We provide new theoretical results demonstrating that low-frequency eigenpairs reliably capture the geometry of the underlying manifold under a constant noise level, up to a dimension independent threshold $O(r^{-2})$, where $r$ is the noise amplitude. Our results rely on a decomposition of the manifold Laplacian in the Sasaki metric, a technique not used before in this area, to our knowledge. We experimentally validate our theoretical predictions. Additionally, we observe similar robust behavior for other manifold learning algorithms which are not based on computing the Laplacian, namely LTSA and VAE.

## 1. Introduction

In this paper, we revisit embedding algorithms such as Diffusion Maps (1; 2), providing a novel perspective on how these methods allow for effective dimension reduction. A large literature has been developed to study non-linear dimension reduction under the so-called *manifold hypothesis* (3), where samples are assumed to be drawn *from* a low-dimensional manifold. Our goal is to show why these techniques remain effective, even in settings where this assumption is violated, with data having been contaminated with high-dimensional noise.

That diffusion maps is robust to noise has been often ob-served. However, the theoretical understanding of manifold estimation in noise has proved to be slow in surrendering its secrets; even in the face of remarkable tours de force such as the ones we mention below, progress has been in small steps. The following seminal papers have both advanced the knowledge of what is possible and brought to light the informational limitations posed by the presence of noise for various statistical settings. In (4; 5; 6) it was shown that manifold *reconstruction* is possible, and (7) studied the minimax rate of covegence obtaining almost tight upper and lower bounds of $n^{-\frac{2}{d+2}}$ for the Hausdorff error (where $d$ is the manifold intrinsic dimension). However, these, as well as (8), consider reconstruction typically in *ambient space* $\mathbb{R}^D$.

For estimating the Laplace-Beltrami operator, the state of knowledge is less advanced. We recall that the $m$ principal eigenfunctions of $\Delta_{\mathcal{M}}$ provide an embedding of a manifold in $m \ll D$ dimensions, by the well known Diffusion Maps (DM) / Laplacian Eigenmaps algorithm and its variants. (9) obtains consistency and rates for the problem of Laplacian estimation, albeit with the assumption that the noise ampli-tude decays to 0 with the sample size $n$. For non-vanishing noise, (10) assume the noise is subgaussian and decaying to 0 with $n$ and $D$ in every direction, but not in $L_2$ norm[1]

**Contributions** Our paper studies the spectral convergence of the noisy manifold Laplacian to the noise-free manifold Laplacian under realistic assumptions of fixed ambient di-mension $D$, and fixed noise amplitude $r$.

Under these conditions we firstly find that only the part of the manifold spectrum below the noise spectrum is re-coverable, with error proportional to the noise amplitude $r$. This leads to a sharp threshold phenomenon in the recov-ery of the spectrum of the noiseless manifold's Laplacian $\Delta_{\mathcal{M}}$ from (infinite) noisy data. The recovery threshold is $O(1/r^2)$ for a fixed noise distribution (constant dimension and amplitude), and we discuss how this relates to noise dimensionality in Section 3.1. Eigenvalues and eigenvec-tors above the threshold are essentially irrecoverable, as the

---

[1]Department of Statistics, University of Washington, Seat-tle, USA [2]Computer Science and Engineering, University of Washington, Seattle, USA. Correspondence to: Alex Kokot <akokot@uw.edu>.

*Proceedings of the $42^{nd}$ International Conference on Machine Learning*, Vancouver, Canada. PMLR 267, 2025. Copyright 2025 by the author(s).

---

[1]Similarly, (5) also rely on $D$ being very large to obtain error bounds (for example $n \sim 10^{10^{d^2}}$ in the second paper), but with exponential sample sizes (in the inverse tolerance and $d$) they succeed to obtain errors that decrease with $n$.

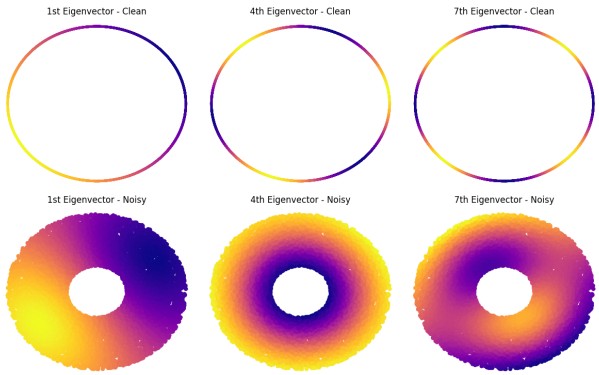

Figure 1: Random walk Laplacian eigenvectors overlaid on samples from the circle ("clean" data) and annulus ("noisy" data). The eigenfunctions of the circle exhibit typical spherical harmonic structure. On the annulus the harmonic eigenfunction of the clean manifold is accurately preserved for the first non-trivial eigenvalue; the second in the display captures orthogonal variation due to noise, and the third has a scrambled (noise+manifold) signal.

geometry is dominated by the noise.

Secondly, in our approach, we leverage properties of the *Sasaki* metric (11), which to our knowledge we are the first to do in this literature. This helps us elegantly disentangle the manifold and noise components of the data distribution, decomposing the Laplacian into a *horizontal* operator along the manifold $\mathcal{M}$ and a *vertical* operator normal to it.

Thirdly, we conduct experiments validating this threshold phenomenon and demonstrating the robustness of low-frequency eigenpairs to noise on real and synthetic datasets. Finally, we compare with two other manifold learning algorithms that do not rely on the Laplacian—Variational Autoencoders (VAEs) (12) and Local Tangent Space Alignment (LTSA) (13)-—and observe similar behavior.

## 2. Notation and Background in Manifold Geometry

Here we succintly introduce the Riemannian geometric concepts used throughout the paper; see (14) for an introduction to Riemannian geometry, which formally describes the objects listed below. For an embedded, oriented $d$ dimensional manifold $\mathcal{M} \subseteq \mathbb{R}^D$, define the tubular neighborhood of radius $r$ about $\mathcal{M}$, $T_r(\mathcal{M}) := \{x \in \mathbb{R}^D : \exists y \in \mathcal{M}, \|x - y\| < r\}$. We will assume $\mathcal{M}$ is at least $C^3$. On occasion where our results apply to more generic topological structures, we refer to $\mathcal{X}$ as a bounded domain with smooth boundary.

The *reach* $\tau$ of $\mathcal{M}$ is defined to be the largest $r$ such that all $x$ in $T_r(\mathcal{M})$ have a unique nearest member of $\mathcal{M}$ w.r.t.

the induced Euclidean metric on $\mathbb{R}^D$. We assume that $\mathcal{M}$ is compact, with $\tau > 0$. Without loss of generality we take $\tau > 1$ by appropriately rescaling our coordinates, and we assume our generative distribution $\mu$ is supported on $T_r(\mathcal{M})$. We can identify the tubular neighborhood with tuples $(x, v) \in \mathcal{M} \times B_r^{D-d}(x)$; each $y \in T_r(\mathcal{M})$ can be uniquely identified by the nearest point $\pi y \in \mathcal{M}$ and the deviation from this point $y - \pi y$. More generally, such pairs $(x, v)$ can be identified with elements of the normal bundle $N\mathcal{M} = \sqcup_{x \in \mathcal{M}} N_x \mathcal{M}$, $N_x\mathcal{M}$ the normal space at $x$, via the exponential map $\exp(x, v)$. For a map $\pi$, we denote its pullback by $\pi^*$ and pushforward by $\pi_*$.

We equip the normal bundle with a *Riemannian metric*, a positive definite bilinear form $g_{(x,v)} : T_{(x,v)}N\mathcal{M} \times T_{(x,v)}N\mathcal{M}$. We will often express $g$ as an inner-product $\langle \cdot, \cdot \rangle$, suppressing the point-wise dependency. A metric on an oriented manifold determines a unique volume form $d\mu_g$.

**Laplacian operators** $\quad \Delta$ denotes the Laplace-Beltrami operator (Laplacian) (15), and $\Delta_\mu$ the *weighted* Laplacian $\Delta_\mu = \Delta + \langle \nabla \log d\mu, \nabla \cdot \rangle$ (16) (we may also denote it $\Delta_p$, with $p$ the density function, again w.r.t the induced Euclidean metric from $\mathbb{R}^D$), with both operators implicitly taken with respect to the induced metric. For corresponding operators in the Sasaki metric (introduced below), we adorn them with a $\sim$, i.e. $\tilde{\Delta}$. When referring to an operator with regard to a specific space, we may denote this with a subscript, i.e. $\Delta_\mathcal{M}$. We consider these operators on their respective Neumann sobolev spaces, denoted $H^2(\mathcal{X}) := H^2(\mathcal{X}, \mu, g)$, for $\mathcal{X}$ the space of interest ($\mathcal{M}$, $T_r(\mathcal{M})$, etc.). When referring to their spectral data, we order the eigenpairs $(\lambda_i, \phi_i)$ such that $\lambda_i \leq \lambda_j$ for $i \leq j$. We often use functional notation to refer to the operator they correspond to, i.e. $\lambda_i(\Delta)$ being the $i$th induced Laplacian eigenvalue. Denote $P_\lambda(\cdot)$ as the projection onto the eigenspaces with eigenvalue $< \lambda$, or alternatively $P_K(\cdot)$ the span of the first $K$ eigenfunctions.

### 2.1. The Sasaki Metric

Our presentation follows (17, Chapter 2.4).

#### 2.1.1. HORIZONTAL AND VERTICAL SPACES

To understand why the induced metric does not adequately capture the desired product structure enabled by the Sasaki metric, consider the annulus displayed in Figure 2. Notice that displacement orthogonally to the circle agrees between the two metrics; lengths and geodesics in one correspond exactly to the other. However, if we consider displacement in the tangent or horizontal direction, the same cannot be said. Visibly, the geodesics of the induced metric follow the euclidean straight lines when possible, and this is particularly problematic when we seek to identify the tangents and orthogonals across different points of this set. Ideally, we

would like to identify each point in the tube with its nearest counterpart along the circle, and this idea extends to when we consider directional differentiation. If at one point on the annulus we had an operator corresponding to orthogonal differentiation and we sought to replicate this at another point by slowly varying this operation across the domain (i.e. parallel transportation) then it would be desirable if at each point along the way the orthogonality to the sphere was maintained. Thus this requires the metric to adjust for the effect of curvature as our path moves closer and further from the inner-most radius, as well as along the manifold if it is not of constant curvature.

Fundamental to this problem is an appropriate identification of a "horizontal space", a subspace of the tangent at a generic point on the tube that corresponds to the tangent of the underlying manifold. Precisely, we represent the tubular neighborhood with a neighborhood of the 0 section of the normal bundle $N\mathcal{M}$ via the exponential map. Taking local coordinates $(x,v)$, $x \in \mathcal{M}, v \in N_x\mathcal{M} \cong \mathbb{R}^{D-d}$ we get a coordinate basis of the tangent space $T_{(x,v)}N\mathcal{M}, \partial/\partial x_1, \ldots, \partial/\partial x_d; \partial/\partial v_1, \ldots, \partial/\partial v_{D-d}$. As indicated above, it would **not** be appropriate to equate the horizontal to the seemingly obvious basis $\partial/\partial x_i$. We must make a selection dependent on the curvature of $\mathcal{M}$ as well as the displacement at $v$. Let us compute how movement in the coordinate directions effects the orthogonal $v$ when moving along the manifold. Take a curve $\gamma_t$ in $\mathcal{M}$ that is incident to $x$ at $t = 0$ with tangent velocity $\alpha := \sum_i \alpha_i \partial/\partial x_i$. Let $u(t)$ be the section of $N\mathcal{M}$ along $\gamma_t$ that is the parallel translation of $v$. Thus $u(t)$ corresponds to a curve in $N\mathcal{M}$, $u(t) = (x_1(t), \ldots, x_d(t), v_1(t), \ldots, v_{D-d}(t))$, and we can immediately compute $\dot{x}_i(0) = \alpha_i$. In our coordinate chart, by construction we also have

$$0 = (\nabla^\perp_\alpha v)^i = \dot{v}_i(0) + \sum_{j,k} \Gamma^i_{j,k} \dot{x}_j(0) v_k(0)$$
$$= \dot{v}_i(0) + \sum_{j,k} \Gamma^i_{j,k} \alpha_j v_k,$$

and thus $\dot{v}_i(0) = -\sum_{j,k} \Gamma^i_{j,k} \alpha_j v_k$, where $\Gamma^i_{j,k}$ are the Christoffel symbols (14) of the induced connection $\nabla^\perp$ on $N\mathcal{M}$. This leads us to a natural definition of the horizontal space as $H_{(x,v)} := \{X : X = \sum_i \alpha_i \partial/\partial x_i - \sum_{j,k} \Gamma^i_{j,k} \alpha_j v_k \partial/\partial v_i\}$, exactly the kernel of the connection map $K^\perp : T_{(x,v)}N\mathcal{M} \to N_x\mathcal{M}, K^\perp(\xi) := K^\perp_{(x,v)}(\xi) := \sum_{i=1}^{D-d}(\xi_{d+i} + \sum_{j,k\leq d} \Gamma^i_{jk} v_j \xi_k) \partial/\partial v_i$.

As suggested earlier, we would like the orthogonal fibers in the Sasaki tube geometry to be isometric to those of the induced metric, Euclidean balls. This corresponds to the identification of the "vertical spaces" as the tangents to the orthogonal subspaces, spanned by $\partial/\partial v_i$, $V_{(x,v)}$. This space is the kernel of $\pi_*$.

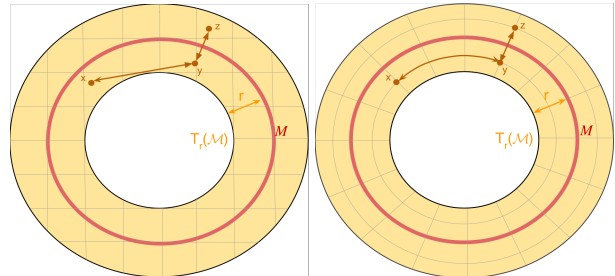

Figure 2: Visualization of the induced(left) and Sasaki(right) metrics on $T_r(S^1)$. The grids represent orthogonal coordinates and the brown lines are geodesics in the respective metrics.

### 2.1.2. METRIC DECOMPOSITION

The Sasaki metric is defined to be

$$\langle u, u' \rangle_{Sa} = \langle \pi_* u, \pi_* u' \rangle_{T_x\mathcal{M}} + \langle K^\perp u, K^\perp u' \rangle_{N_x\mathcal{M}},$$

orthogonalizing $T_{(x,v)}N\mathcal{M} = H_{(x,v)} \oplus V_{(x,v)}$, and independently preserving horizontal and vertical displacement (18).

For weighted Laplacians on $T_r(\mathcal{M})$, we will assume that the respective densities $d\mu$ (w.r.t. the ambient measure) factors along the horizontal and vertical spaces, $d\mu(x, v) = d\mu_{\text{Sa}} p(x) q(v)$, identifying $q$ with a fixed function on the canonical fiber $B_r^{D-d}$. In other words, as made precise in Lemma A.1, we can generate samples from $\mu$ by sampling the manifold, then adding orthogonal noise that only depends on the selected point via the direction of the orthogonal. For example, $X$ generated by sampling $M$ supported on $\mathcal{M}$ then adding isotropic Gaussian noise in the direction of the orthogonal meets this requirement. This noise structure is typical in the manifold learning literature ((7; 19), etc.). We further assume that this orthogonal noise has compact support equal to a ball of radius less than the reach.

### 2.2. The Sasaki Laplacian for $T_r(\mathcal{M})$

The primary utility of the Sasaki metric is the spectral structure it induces. Consider the noisy circle (annulus), which we equip with the product metric $S^1 \times (a, b)$. If we take the Neumann (insulated) boundary condition, then we see by a separation of variables argument that the Laplace-Beltrami operator decomposes as $\Delta = \Delta_{S^1} \oplus \Delta_{(a,b)} =: \Delta_{S^1} \oplus \Delta_r$, where $r$ is the width $|b - a|$. As $r \to 0$, one can compute that the minimal non-trivial eigenvalue of $\Delta_r$ grows as $r^{-2}$, with the consequence being that any eigenfunction with smaller eigenvalue must be constant orthogonally to the circle.

The above properties hold in general for the Sasaki metric of the tubular manifold $T_r(\mathcal{M})$, as shown in Proposition 2.1

below. Before we state the proposition, we introduce the following technical artifice that will be useful throughout the paper.

Rather than considering differential operators with different domains $T_r(\mathcal{M})$, for $r \in (0,1)$, we fix $T_1(\mathcal{M})$ as a reference space. Then we identify the Sobolev spaces $H^2(T_1(\mathcal{M}))$ with $H^2(T_r(\mathcal{M}))$ via $f \to f \circ \sigma_r$, where $\sigma_r(x,v) := \exp(x, v/r) = (x, v/r)$ is called a *rescaling map*. In other words, to study the spectra of the contracted operator $\tilde{\Delta}_{\mu,r} := \Delta_{\sigma_{r*}\mu}$, it suffices to study its conjugate $\tilde{\Lambda}_{\mu,r} := \sigma_r^{*-1}\tilde{\Delta}_{\mu,r}\sigma_r^*$, the weighted Laplacian in the metric $\langle \cdot, \cdot \rangle_{\mathrm{Sa},r} := \langle \sigma_r^* \cdot, \sigma_r^* \cdot \rangle_{\mathrm{Sa}}$, which has the same spectrum as $\tilde{\Delta}_{\mu,r}$.

We verify properties of a horizontal-vertical decomposition of the weighted Laplacian analogous to the unweighted case previously presented in (18; 20; 21).

**Lemma 2.1.** *Let $\mathcal{M}, T_r(\mathcal{M})$, $\mu$ on $T_r(\mathcal{M})$ factorizing w.r.t. the horizontal and vertical spaces, and Laplacians $\Delta_{\mathcal{M},p}, \tilde{\Delta}_{\mu,r}$ as defined above. Then the following hold.*

1. *$\tilde{\Lambda}_{\mu,r}$ has discrete spectrum with finite multiplicity eigenvalues, and decomposes as*

$$\tilde{\Delta}_{\mu,r} = \Delta_{H,p} + \frac{1}{r^2}\Delta_{V,q}.$$

   *for some operators $\Delta_{H,p}, \Delta_{V,q}$ such that $\Delta_{H,p}(f) \circ \pi = \Delta_{\mathcal{M},p}\pi^* f$, $\Delta_{V,q}(f)(x,v) = \Delta_{B(0,1)}(f|_{\pi_x^{-1}})(v) + \langle \nabla \log q, \nabla f|_{\pi^{-1}(x)}(v) \rangle$,*

2. *The operators $\tilde{\Lambda}_{\mu,r}$, $\Delta_{V,q}$, and $\Delta_{H,p}$ are non-negative and self-adjoint.*

3. *$\tilde{\Lambda}_{\mu,r}$, $\Delta_{H,p}$, and $\Delta_{V,q}$ commute pairwise; hence, there is a common orthonormal basis $\{\phi_i\}_{i=1}^\infty \subset L^2(T_1(\mathcal{M}), \mu_{\mathrm{Sa}})$ of smooth eigenfunctions such that*

$$\Delta_{H,p}\phi_i = \lambda_i^H \phi_i, \quad \Delta_{V,q}\phi_i = \lambda_i^V \phi_i,$$

$$\tilde{\Lambda}_{\mu,r}\phi_i = (\lambda_i^H + \frac{1}{r^2}\lambda_i^V)\phi_i.$$

   *These eigenfunctions and spectra $\lambda_i^H, \lambda_i^V$ are invariant for $0 < r < \tau$*

4. *For $\lambda_i(\tilde{\Lambda}_{\mu,r}) \leq \lambda_1^V/r^2$, $\phi_i(\tilde{\Lambda}_{\mu,r}) = \phi_i(\Delta_p) \circ \pi$ and $\lambda_i(\tilde{\Lambda}_{\mu,r}) = \lambda_i(\Delta_p)$.*

Proposition 2.1 states that the spectrum and eigenfunctions of the Sasaki Laplacian of the noisy manifold $T_r(\mathcal{M})$ recover the noise free coresponding quantities of $\Delta_{\mathcal{M}}$ *below the noise dependent threshold $\propto r^{-2}$*. This supports the intuition that the principal eigenspaces of a manifold Laplacian are robust to noise, and, moreover, that the robustness extends to larger eigenvalues when the noise amplitude $r$ decreases.

# 3. Spectral Analysis of Noisy Manifold Laplacian

## 3.1. The Low Spectrum of the Noisy Manifold Laplacian

As we saw in the case of the Sasaki metric, the low-order spectrum of the Laplacian bears close resemblance to that of the underlying manifold. We extend these results to the induced metric Laplacian via a perturbation analysis.

**Theorem 3.1** (Spectral perturbation for small eigenvalues)**.** *Let $\Delta$ be the Laplacian on $T_r(\mathcal{M})$, $r < 1 < \tau$, with measure $\mu$; in the weighted case take $\mu$ factoring w.r.t horizontal and vertical spaces, $\mu_r := \sigma_r^*\mu$, otherwise let $\mu_r$ be the ambient measure. Let $\Delta_{\mathcal{M}}$ be the corresponding operator on $\mathcal{M}$, in the weighted case with respect to the density $p$. Let $\lambda_k(\Delta_{\mathcal{M}}) < \lambda_1^V/r^2$.*

*There exists a constant $C$ depending on $\mathcal{M}$ such that*

$$|\lambda_k(\Delta_{\mathcal{M}}) - \lambda_k(\Delta)| \leq 8\lambda_k Cr.$$

*Further, for $\lambda < \lambda_1^V/r^2$, there exists a constant $C_\lambda$ such that*

$$\|\pi^* P_\lambda(\Delta_{\mathcal{M}}) - P_\lambda(\Delta)\| \leq 16C_\lambda Cr,$$

*in particular, for any $\lambda_k(\Delta_{\mathcal{M}}) \leq \lambda$ simple, $\phi_k(\tilde{\Lambda}_{\mu,r}), \phi_k(\Lambda_{\mu,r})$ unit, there exists $C'$ depending on $\lambda, \mathcal{M}$ such that*

$$\|\pi^*\phi_k(\Delta_{\mathcal{M}}) - \phi_k(\Delta)\|_{L^2(\mu_r)} \leq C'\lambda_k(\Delta_{\mathcal{M}})r,$$

**Corollary 3.2.** *Let $\mathcal{P}(\mu)$ denote the Poincaré constant of a distribution $\mu$. With the above notation, the Poincaré constant $\mathcal{P}(\mu_r) = \mathcal{P}(\pi_*\mu) + Cr$ with $C$ only depending on $\mathcal{M}, p$.*

In other words, in the low-frequency regime, the Neumann eigenfunctions of $\Delta_{T_r(\mathcal{M})}$ agree up to small error with constant orthogonal extension of the eigenfunctions of the base manifold, $f \circ \pi$. Thus, procedures like diffusion maps, when truncated to a small basis, perform dimension reduction by implicitly denoising the data. In the high frequency regime, such a recovery is hindered not only by the magnitude of the perturbation, which is proportionate to the corresponding eigenvalue, but also by the presence of non-trivial "noise" eigenfunctions. In fact, as the noise dimension $D - d$ is typically much larger than the manifold $d$, even in the Sasaki metric the number of eigenfunctions with no orthogonal dependence is negligible in the limit. This, along with the shrinking spectral gaps (of the resolvent) and, as previously noted, increasing perturbation size, lead to unrecoverable manifold data in the continuum.

We now discuss byproducts of our result. As we see in Corollary 3.2, this spectral approximation translates to guarantees on mixing times (22) and geometric inequalities (23) that closely relate the concentrated tube to the underlying

manifold. Our bound also suggests a similar consistency for noise with increasing dimension as verified for empirical Laplacian estimation in (10). In typical examples $\lambda_1^V$ increases with the dimensionality of the noise, thus yielding the result if coupled with a verification that the corresponding perturbation magnitude also decreases. While we see this empirically for orthogonally uniform noise in Figures 5 and 11, we conjecture that such consistency can be verified under a generic log-concavity assumption.

Our analysis is geared toward the study of very regular noise structures, and this is more restrictive than one might expect for real data. However, in practice, these same spectral phenomena are observed in far more irregular sampling settings so long as the measure is sufficiently concentrated about a low-dimensional structure. We sketch how our approach can be extended to deliver theoretical guarantees in such cases. For $\mu$ with smoothly varying tube widths/orthogonal densities, one can construct a natural diffeomorphism $\psi$ such that the support of $\psi_* \mu_r$ is a tube manifold, and $\psi_* \mu_r$ factors w.r.t horizontal and vertical spaces. This amounts to stretching or contracting the measure smoothly to achieve the desired structure. This is an isometry if we equip the resulting tube with the metric $(\psi^{-1})^* g$, for $g$ the original Euclidean metric. Thus, the results of Appendix 3.1 are immediately applicable if we can verify that under rescaling this metric sufficiently approximates the Sasaki. We expect this to be similar to the verification for the induced metric, but we leave this to future work. In the following section we prove generic spectral growth bounds applicable to irregular domains.

### 3.2. Multi-scale Spectral Growth and a Weyl law

As a retrospective, we note that aspects of the spectral phenomenon presented in Section 3.1 are present in more general settings. Tubular neighborhoods are a particular example of multi-scale distributions, distributions that, in an appropriate sense, have dimension that changes depending on the scale which one localizes at. This is made precise via the packing number $\mathrm{Pack}(T_r(\mathcal{M}), \varepsilon) = \mathrm{Pack}(T_r(\mathcal{M}), \varepsilon, \|\cdot\|_\infty)$, the maximum number of disjoint cubes with side length $\varepsilon$ and center in $T_r(\mathcal{M})$. Packing and the related covering number appear ubiquitously in statistical learning when describing the complexity of a space, and have been tied to convergence rates of other geometric quantities such as the Wasserstein distance (24).

For $\varepsilon$ relatively large compared to $r$, $\mathrm{Pack}(T_r(\mathcal{M}), \varepsilon)$ grows at the intrinsic rate $\varepsilon^{-d}$, but as $\varepsilon$ dives below the noise threshold we get the extrinsic $\varepsilon^{-D}$. Not coincidentally, this is likewise seen in the growth rate of the spectra above and below $\lambda_1^V / r^{-2}$, measured by the *counting function*

$$N(\lambda) := |\{i : \lambda_i(\Delta_{\mathcal{X}}) \leq \lambda\}|.$$

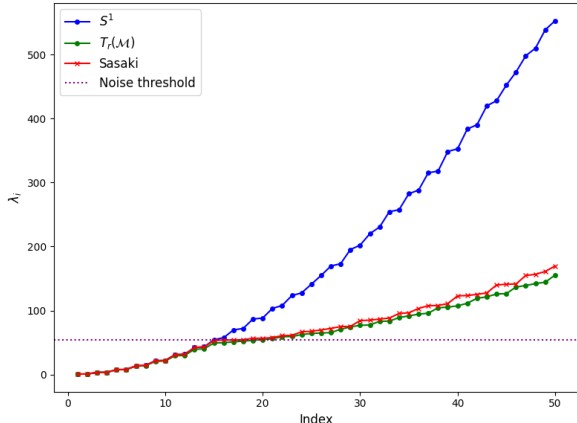

Figure 3: Spectral growth of the circle and annulus from Figure 1 in the induced and Sasaki metrics, eigenvalues empirically estimated.

As shown in Figure 3, below this threshold, the tubular neighborhood has eigenvalue growth reflecting that of the base manifold. Above the threshold, the growth is slower, that is, the counting function $N(\lambda)$ is larger, reflecting its asymptotic behavior as an extrinsic dimensional domain in $\mathbb{R}^D$. The relationship between Laplacian spectra and coverings is classical, dating back to the original proof of the Weyl Law (25). Several recent papers (26; 27) have bounded the spectral growth of the Neumann Laplacian from this perspective. Leveraging their approaches, we precisely quantify the low-order spectral growth, characterizing it in terms of packings. Note that, in the limit, the typical Weyl law verifies the extrinsic dimensional dependence $\lambda_i \sim i^{2/D}$.

**Theorem 3.3.** *Let $\mathcal{X}$ be a bounded domain with smooth boundary, and define*

$$C(k, \varepsilon) := \sup_{x \in \mathcal{X}} \mathrm{Pack}(B_{\mathcal{X}}(x, k\varepsilon), \varepsilon),$$

$$R(\varepsilon) := \frac{\inf_{x \in \mathcal{X}} \mu(B(x, \varepsilon))}{\sup_{x \in \mathcal{X}} \mu(B(x, \varepsilon))}.$$

*Then,*

$$N(C(4, \varepsilon)\varepsilon^{-2}) \leq \mathrm{Pack}(\mathcal{X}, \varepsilon)/(4C^2(4, \varepsilon))R(\varepsilon).$$

*If, in addition, $\mathcal{X}$ is convex or has empty boundary, then there exists $C'$ depending on $\mathcal{X}$ such that*

$$\mathrm{Pack}(\mathcal{X}, \varepsilon) \leq N(C'\varepsilon^{-2}/C(2, \varepsilon)).$$

For our tubular neighborhood model $T_r(\mathcal{M})$, the first of these inequalities implies that eigenvalue frequency *below noise level* reflects the base manifold.

**Corollary 3.4** (Local Weyl law for $\Delta_{T_r(\mathcal{M})}$). *Let $\Delta_{T_r(\mathcal{M})}$ be defined as in the previous sections. Then, for $\varepsilon > 2r$,*

*there exists a constant $C > 0$ depending only on $\mathcal{M}$, such that $C(4, \varepsilon)\varepsilon^{-2} \leq \lambda_{C\varepsilon^{-d}}$.*

In the results above, we leave many of the bounds expressed in full detail, even though many of the quantities could be collapsed down to a constant. For example, $C(4, \varepsilon)$ is simply the maximal number of $\varepsilon$ cubes that can be packed into a $4\varepsilon$ $\mathcal{X}$ cube. That this is no larger than $4^D$ is immediate, however it is much smaller ($\approx 4^d$ for the tube) for $\varepsilon$ not too small, which is frequently the case of interest. Similarly, $R(\varepsilon)$ tends to 1 for $\varepsilon$ large and $1/2$ as $\varepsilon \to 0$, with a global bound depending on how curved the boundary of the domain is.

That the tube is multi-scale also plays an essential role in the estimation of its spectral data. At face value, this problem suffers from the curse of dimensionality, with convergence rates depending exponentially on the ambient dimension $D$. However, the approximation quality of Diffusion Maps is proportionate to the $W_\infty$ distance between the empirical distribution $\mu_n$ and $\mu$ (9). For $\mu$ supported on $T_r(\mathcal{M})$, a simple application of the triangle inequality yields $W_\infty(\mu_n, \mu) = O(W_\infty(\pi_*\mu_n, \pi_*\mu) + r)$. Thus, achieving estimation error at the order of the bias is an intrinsic dimensional problem.

# 4. Experiments

## 4.1. Data

We experimentally verify the extent to which the approximation in Theorem 3.1 holds for both synthetic and real datasets. Here we briefly describe the data used in our experiments, while more details can be found in Appendix D.

**Synthetic Data**: For our synthetic experiments we sample data from the following manifolds: $S^d$, the S-Shape, and the Bottle (a more complicated manifold described in Figure 7) to which we add noise uniformly distributed on a ball of radius $r$ in the normal space to the manifold.

**Real Data**: For our real data experiments we use Molecular Dynamic Simulations (MDS) datasets consisting of atomic configurations from three molecules: Toluene(Tol), Malonaldehyde(Mal), and Ethanol(Eth) from (28). These simulations require massive compute power, are extremely precise, and generate atomic configurations that exhibit non-linear, multiscale, non-i.i.d. noise, as well as complex topology and geometry, often lying near low-dimensional manifolds (29). We perform SVD to reduce the ambient dimension to $D = 50$ and add noise uniformly distributed on a ball of radius $r$ in the ambient space.

## 4.2. Validation of Theoretical Results

In this section we empirically validate the predictions of Theorem 3.1. For this purpose, we estimate the Laplace operators $\Delta_\mathcal{M}$ and $\Delta_{T_r(\mathcal{M})}$ and their spectral decomposition from data using the renormalized Diffusion Maps Laplacian (1). We do so for all manifolds described in Section 4.1 and for many tubular neighborhood radii $r \in (0, \tau)$ and dimensions of the normal space $d_n = D - d$.

First, we use the estimated eigenpairs to confirm the robustness to noise (both $d_n$ and $r$) of the low-frequency eigenpairs of $\Delta_{T_r(\mathcal{M})}$ below the *noise threshold* $\frac{\lambda_1^V}{r^2}$. We do this quantitatively by verifying that the error between the low-spectrum eigenspaces (represented by their projectors) and their noiseless counterparts increases linearly with $r$. Furthermore, we notice a mild decrease in error as $D$ gets larger, as argued in (10). Qualitatively, we observe that low-frequency $\lambda_i(\Delta_{T_r(\mathcal{M})})$ and $\phi_i(\Delta_{T_r(\mathcal{M})})$ remain stable as $d_n$ is varied and as long as $\lambda_i(\Delta_{T_r(\mathcal{M})}) < \frac{\lambda_1^V}{r^2}$.

Second, we show that, as the theory suggests, eigenvectors with $\lambda_i(\Delta_{T_r(\mathcal{M})}) > \frac{\lambda_1^V}{r^2}$ are corrupted by noise, the noise threshold acting as a delimiter between the noiseless and the noisy eigenpairs. Quantitatively, we verify this prediction by computing the empirical correlations between $\phi_i(\Delta_{T_r(\mathcal{M})})$ and $\phi_j(\Delta_\mathcal{M})$ which suffer a sharp decrease for eigenvectors above the noise threshold, a behavior that is independent of sample size. Qualitatively, we observe that the values of $\lambda_i(\Delta_{T_r(\mathcal{M})})$ are perturbed as they approach or go above $\frac{\lambda_1^V}{r^2}$ and that the corresponding eigenvectors are no longer correlated with the coordinate space of $\mathcal{M}$.

The experiments above are visually depicted in Figure 4 and elaborated upon in the Appendix E.

# 5. Other Manifold Learning Algorithms

Although our main theoretical results pertain only to the Laplace operator, we observe the same behavior outlined in Theorem 3.1 and verified in our experiments for VAE (12) and LTSA (13), two manifold learning algorithms not based on Laplacian decomposition.

As opposed to DM, we have no natural way to evaluate whether a VAE latent coordinate or LTSA embedding dimension $\phi_i$ is low or high frequency. For this purpose we use the regression procedure outlined in Section E.2. Briefly, for the synthetic data we know the noise/normal space coordinates that generated each data point. Using these as input, we fit polynomial regression models that attempt to fit the $\phi_i$'s as targets. If the model is able to predict the embedding coordinates from the noise coordinates, we deem this embedding coordinate to be 'high frequency' as it encodes noise information.

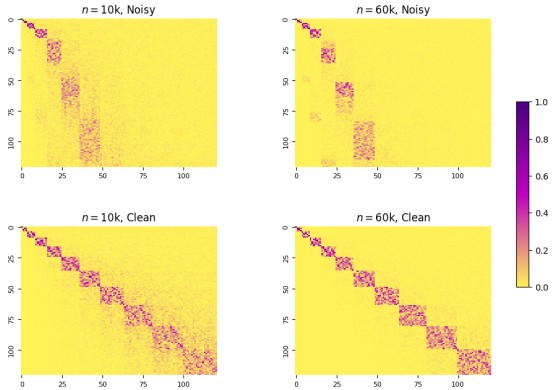

(a) Heatmaps of the empirical correlations between $\phi_i(\Delta_{T_r(\mathcal{M})})$ (rows) and $\phi_j(\Delta_{\mathcal{M}})$ (columns) on $S^2$. Notice the sharp decrease in correlation once $\lambda_i(\Delta_{T_r(\mathcal{M})}) > \lambda_1^V/r^2$.

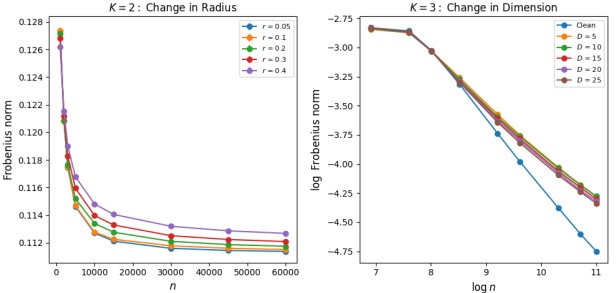

(b) Error between the projectors onto the $K$-th eigenspace of $\Delta_{\mathcal{M}}$ (computed analytically) and $\Delta_{T_r(\mathcal{M})}$ (estimated) on $S^2$. Notice the linear increase in error as $r$ increases (left) and the mild decrease in error as $D$ increases (right).

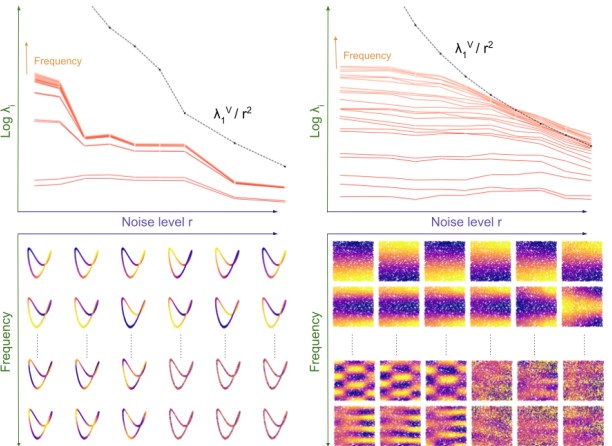

(c) Qualitative validation of our results on two manifolds: the Toluene molecule(left) and the Bottle(right) with $d_n = 3$. In the top plots, each path corresponds to one $\lambda_i(\Delta_{T_r(\mathcal{M})})$ which are almost constant until $\lambda_1^V/r^2$(the black line) approaches them. In the bottom plots, we color the coordinate spaces of the manifolds by low-frequency (top) and high-frequency (bottom) $\phi_i(\Delta_{T_r(\mathcal{M})})$. Notice the stability of the former as $r$ increases and the sudden corruption of the latter once the noise threshold is passed.

Figure 4: Summary of the experiments validating our theoretical results

First, we observe that VAEs will learn a set of embedding coordinates $\phi_i$ that are stable w.r.t. to both $r$ and $d_n$ and which are indispensable for reconstruction. However, if the latent space has more capacity than that, then the VAEs will use the extra coordinates to encode noise. Second, the embeddings computed by LTSA, a classical manifold learning algorithm that does not rely on Laplacian estimation, behave similarly to the eigenvectors $\phi_i(\Delta_{T_r(\mathcal{M})})$ we estimated in Section 4.2 using Diffusion Maps.

Our results are summarized in Figure 5 and expanded in Appendix F where we fully describe the models and hyperparameters used.

# 6. Discussion and Related Work

Embedding manifold data in low dimensions by eigenvectors has been one of the most successful directions in non-linear dimension reduction (13; 30) both practically and from the point of view of the theoretical understanding it allows. Among these algorithms, the DM algorithm of (1; 2) using the eigenvectors of (a matrix estimating) the Laplace-Beltrami operator $\Delta_{\mathcal{M}}$ of the manifold has been a central subject of study, but the practical success of spectral methods is due to their robustness on real data that rarely satisfy the manifold hypothesis.

In this paper, we provide a new perspective on why, and to what extent, this robustness is to be expected. We study the relation between the ideal $\Delta_{\mathcal{M}}$ and the Laplace-Beltrami operator $\Delta$ of $T_r(\mathcal{M})$, a noisy version of $\mathcal{M}$.

We show (in Theorem 3.1) that indeed the eigendecomposition of the Neumann Laplacian in the perturbed domain differs from that of the intrinsic manifold $\mathcal{M}$ by a mild error in the low-frequency regime, thus corroborating the traditional wisdom that eigenfunctions associated to small eigenvalues are stable. But our results in Sections 3.1 and 2.2 also strongly suggest that the recovery is limited to the part of the $\Delta_{\mathcal{M}}$ spectrum below a threshold $O(r^{-2})$, representing the first non-trivial noise eigenvalue, after which recovery becomes impossible, due to the high density of the noise spectrum above this threshold.

Furthermore, in Section 3.2, we present more general results and intuitions on the spectral density for a larger class of geometric objects in noise, as we relate eigenvalue growth rate to packing numbers. These results allow for general noise structures, tying into our sketch of how our noise assumptions can be loosened presented in Section 3.1.

The study of manifold estimation in noise is more than a decade old (3; 4; 5; 10; 7; 31), and we adopt the standard assumptions of "tubular" i.i.d. noise. But in contrast with previous works that have studied the impact of noise in related geometric (10) and kernel based (31) algorithms, our

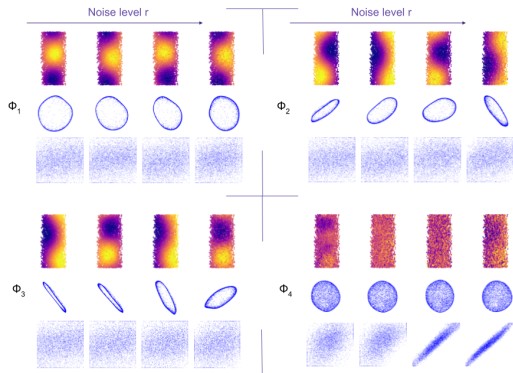

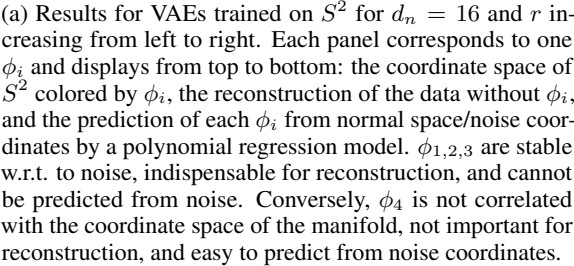

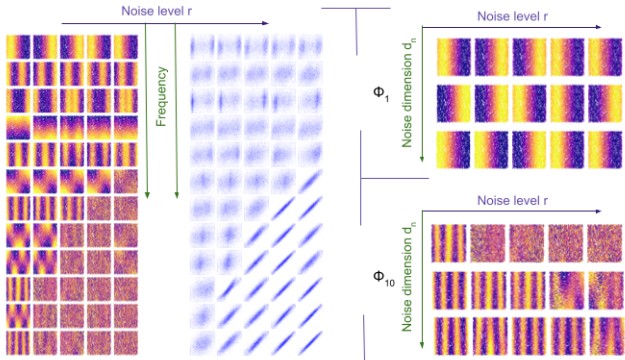

(a) Results for VAEs trained on $S^2$ for $d_n = 16$ and $r$ increasing from left to right. Each panel corresponds to one $\phi_i$ and displays from top to bottom: the coordinate space of $S^2$ colored by $\phi_i$, the reconstruction of the data without $\phi_i$, and the prediction of each $\phi_i$ from normal space/noise coordinates by a polynomial regression model. $\phi_{1,2,3}$ are stable w.r.t. to noise, indispensable for reconstruction, and cannot be predicted from noise. Conversely, $\phi_4$ is not correlated with the coordinate space of the manifold, not important for reconstruction, and easy to predict from noise coordinates.

(b) Results for LTSA for the S-Shape. **Left**: Stability w.r.t to $r$ of the low-frequency embeddings and the threshold effect exhibited by the high-frequency $\phi_i$ as they become drowned by noise. We color the coordinate space of the manifold by $\phi_i$ as $r$ increases from left to right. In the next panel, we use a polynomial regression model to predict each $\phi_i$ from the known normal space/noise coordinates. When we are able to do so, we can conclude that the corresponding $\phi_i$ represents encodes. **Right**: Stability of $\phi_1$ w.r.t to varying $r$ and $d_n \in \{16, 24, 32\}$(top) vs the lack thereof of the higher- $\phi_{10}$(bottom).

Figure 5: Summary of results for other manifold learning algorithms.

present work does not require the amplitude and dimensionality of the noise to change with the sample size. Further, this analysis is agnostic to the particular choice of estimator (e.g. (1; 10)), as we show that the continuum Laplace-Beltrami operator itself has desirable robustness properties with regard to its principal eigenvalues and eigenfunctions. To handle the fixed noise assumption, we introduce new techniques, based on the elegant Sasaki metric concept (11).

Our proof technique resembles those employed for Laplacian spectral estimation (9; 32). Central to both arguments is a realization of the near isometry between two spaces, in our case the induced and Sasaki metrics, while former works have emphasized comparisons between empirical and population distributions. Studying the effects of noise on the Laplacian and related heat operators has a rich literature in Differential Geometry. We utilize key techniques from (20; 33).

While we do not currently have a matching lower bound, we anticipate that the bias proportionate to $r$ is the optimal possible, particularly for less regular noise structures. In the tubular setting, it is possible to exploit our strict noise structure to learn $\mathcal{M}$ itself (7), and this can be employed to sharpen the estimates of the intrinsic spectrum and estimate the cut-off for reliable estimation. Note that since our results are *in population*, if they are tight, no procedure will be able to recover the Laplacian eigenfunctions at a better accuracy. Beyond the scope of this submission, we have considered the recoverability of the eigenfunctions and spectrum above the $O(r^{-2})$ threshold. We find strong indications that this

recovery is not possible, based on the density of noise eigenvalues that far exceeds the density of the $\Delta_{\mathcal{M}}$ eigenvalues in the spectrum. We leave this for an upcoming paper.

Other benefits from our results are as follows. First, practically, by verifying this result in the continuum, we provide novel asymptotic guarantees for (biased) spectral recovery even in the presence of noise. The second benefit is a better understanding of how spectral algorithms like DM provide meaningful dimension reduction. In the noiseless case, it is a geometric result that infinite dimensional eigenbases recover the geometry of the domain (34; 35), hence seemingly more eigenvectors would recover more information. However, here we show that with noise, using more eigenvectors will overfit the data by recovering the uninformative geometry of the noise. It is exactly the low dimensional truncated eigenvector bases commonly seen in practice (1; 13; 36) that provide the highest resolution of the underlying manifold. This leads to an interesting tension when compared to learning procedures in supervised settings that rely on infinite Laplacian bases (37), particularly when the signal only depends on the manifold component. In future work, we aim to study this problem in more detail, with the goal of reducing this inefficiency.

What do our results say about the robustness of Diffusion-Maps algorithm in practice? First, note that the error bound $\mathcal{O}(r)$ does imply robust, if inexact recovery in real data. In particular, in typical examples as the noise threshold grows proportionately to the noise dimension, our results allow for noise amplitudes of $r\sqrt{D-d}$ which can be quite large.

Second, in real scenarios, noise is often not full $D-d$ dimensional, but has a multi-scale structure, where the large noise is only in $D' < D$ dimensions. Our results do not depend on $D$, hence they (approximately) cover this case too. Third, if the eigengaps $\lambda_k(\Delta_{\mathcal{M}}) - \lambda_{k-1}(\Delta_{\mathcal{M}})$ are sufficiently small, it is possible that the respective $\lambda_k(\Delta), \lambda_{k-1}(\Delta)$ change order, while the combined eigenspace remains approximately the same. Thus, if we were to use the eigenvectors of $\Delta$ for obtaining an embedding, the selection of $m$ eigenvectors ($m > d$ being the embedding dimension) may vary between noise level and noise instantiations on the same underlying manifold $\mathcal{M}$.

In our theorems, we do not assume that the sampling is uniform; but we make the strong assumption that the sampling distribution $\mu$ factors w.r.t. the horizontal and vertical spaces, in other words, that the noise is orthogonal, has tubular support, and is independent of $x$. If the noise is not isotropic, or if the noise distribution is not uniform (but still independent on $x$), it is easy to see that Theorem 3.1 will continue to hold, albeit with a different threshold of $\lambda_1^V r^{-2}$, where $\lambda_1^V$ is the lowest eigenvalue of the noise distribution, as in Section 2.1. If the noise becomes highly anisotropic, in other words, if the density of noise eigenvalue decreases (for example, if $\lambda_1^V$ has multiplicity 1), then some eigenpairs beyond the noise threshold may be recoverable as well.

Empirically, DM and other spectral algorithms are robust also when the noise varies along the manifold, as can be seen e.g. in the MDS experiments. While our results do not cover this case, we believe that they can be extended, by making the assumption that the noise is approximately i.i.d. In such a case, an additional estimation error will be incurred due to the difference between tubular noise and the actual noise. Furthermore, in future work, we believe we can extend our results to *slowly varying* departures from tubular noise.

Our experiments in Sections 4 confirm that the theoretical results carry over to finite samples, where we observe eigenfunction perturbations on the order of the tube width, as well as perturbation reduction with increased noise dimension. We see this repeatedly in qualitative experiments, where sufficient sample noise and spectrum depth leads to loss of eigenfunction signal. Our results can be combined with those of known empirical estimators of Laplacian eigendata (32; 38), however additional work is needed to confirm these results in the context of non-trivial boundary. For appropriate bandwidth selection for these estimators, the method of (39) can be employed for estimation of the intrinsic dimension, however as indicated by our spectral analysis, data adaptive procedures such as (40) adequately adjust to multi-scale tubular data.

The experiments in Section 5 with LTSA and VAE demonstrate a similar phenomenon. As LTSA is principally a manifold learning algorithm, this close correspondence is not surprising. VAE on the other hand bears no direct relationship to DM, however we observe the same qualitative behavior. Fitting our generative distribution with a low-capacity, low-dimensional embedding, we arrive at a partitioning of the coordinates into manifold components and noise.

## Acknowledgments

This research was carried out mainly in the Statistics Department at University of Washington. A.K. was partially supported by by the Department of Defense (DoD) through the National Defense Science & Engineering Graduate (ND-SEG) Fellowship Program. M.M. gratefully acknowledges the hospitality of the DataShape Group at INRIA Saclay and of the Institute for Mathematical and Statistical Innovation (IMSI) through the long program on "Data-Driven Material Informatics".

## Impact Statement

This paper presents work whose goal is to advance the field of Machine Learning. There are many potential societal consequences of our work, none which we feel must be specifically highlighted here.

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

# Appendix / supplemental material

## A. Extended Background

This section seeks to clarify basic properties of the Sasaki metric and Laplacian (11).

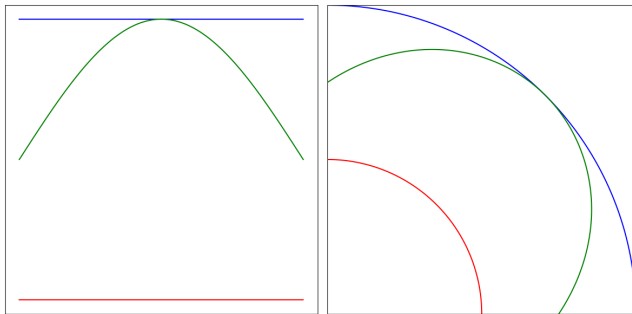

Figure 6: A curves on the annulus displayed in the Sasaki metric (left) compared to the induced (right).

**Example and intuition**   As motivation, consider a circle with noise, an annulus, with inner-radius 0.5 and outer radius 1.5. A natural parameterization for this surface is $(\theta, r) \in \mathbb{R}/2\pi\mathbb{Z} \times (0.5, 1.5)$, the angular and radial coordinates. With this representation, a simple metric structure is the product metric,

$$d((\theta_1, r_1), (\theta_2, r_2)) = \sqrt{\|\theta_1 - \theta_2\|^2_{\mathbb{R}/2\pi\mathbb{Z}} + (r_1 - r_2)^2}.$$

Of course, this structure fails to account for the curvature of the circle, as demonstrated in Figure 2. For example, in the Euclidean geometry, two points on the inner-most circle are comparatively much closer than those on the outer-most, while the Sasaki metric returns the same value in both instances. However, if these radii are close together then these values will only differ negligibly (under an appropriate scaling), and the Sasaki geometry recovers the ambient up to a small perturbation.

**Measure factorization**   When considering manifolds equipped with a measure $\mu$, we characterize below the appropriate densities for our results are applicable.

**Lemma A.1.** *A distribution $\mu$ supported on $T_r(\mathcal{M})$ factors along horizontal and vertical spaces if and only if $X \sim \mu$ is equal in distribution to $M + E_M$ where $M$ is sampled with density $p$ on $\mathcal{M}$ and $E_M$ is supported on the orthogonal to $M$, such that $(\gamma_t)_* E_M \overset{d}{=} E_{\gamma_t(M)}$ for $\gamma_t$ the flow of a basic vector field.*

*Proof.* Let $B_r$ be the ball of radius $r$ in $\mathbb{R}^{D-d}$. By construction, $T_r(\mathcal{M})$ equipped with the Sasaki metric is locally isometric to the product structure $\mathcal{M} \times B_r$. Indeed, we can follow an argument in local coordinates as seen in (18, Theorem 1.5). We take a bundle chart in $TN\mathcal{M}$ with local coordinates $(x, v, \xi, \eta)$. Due to the decomposition $T_{(x,v)}N\mathcal{M} = H_{(x,v)} \oplus V_{(x,v)}$, we adopt a local frame $\phi_1, \ldots, \phi_d, \psi_1, \ldots, \psi_{D-d}$ such that the $\phi_i$ span the horizontal subspace, and the $\psi_i$ the vertical. As these subspaces are orthogonal, we can wlog adapt our coordinate system so that $(x, v)$ are such that $\partial/\partial x_i = \phi_i, \partial/\partial v_i = \psi_i$, and thus our metric is the product metric on $X \times V$. The left factor is isometric to a domain in $\mathcal{M}$ as the Sasaki metric is a Riemannian submersion, and the right factor is isometric to $B_r$ by (18, Proposition 5.2). Classical computations (17, Section 2.4) yield that such coordinates can be constructed about $m \in \mathcal{M}$ via the exponential map of $TN\mathcal{M}$, $(x, v) \rightarrow \exp_{(m,0)}(x + v)$, $x \in H_{m,0} = T_m\mathcal{M}$, $v \in V_{m,0} = N_m\mathcal{M}$, $x, v$ localized appropriately.

Now, let us assume first that the normal bundle of $\mathcal{M}$ is trivial, so that the isometry above was global. In this case, the assumption that $\mu$ factors w.r.t horizontal and vertical spaces can be expressed equivalently as $\mu$ being a product measure on $\mathcal{M} \times B_r$ by the above isometry. In this case the equivalence is trival as flows of basic vector fields leave $B_r$ invariant.

In general, as the flow of $\gamma_t$ from any initial point $m \in \mathcal{M}$ is a compact path, we can cover it by a finite collection of bundle charts, for each of which the claim is verified. Hence, by the compatibility of these charts, the claim holds across the whole path. $\qquad\square$

# B. Spectral Analysis

## B.1. General Properties of the Sasaki Metric. Proof of Proposition 2.1

The proof of Proposition 2.1 consists of a series of steps, each in supporting statements proved separately. See Section 2 for relevant definitions/assumptions.

**Lemma B.1.** *Let $u \in V_{(x,v)}$, then*

$$\langle K^{\perp} u, K^{\perp} \nabla f(x,v) \rangle_{N_x \mathcal{M}} = \langle K^{\perp} u, \nabla f|_{\pi^{-1}(x)}(v) \rangle_{N_x \mathcal{M}}.$$

*For $u \in T_x \mathcal{M}$,*

$$\int_{\pi^{-1}(x)} \langle \tilde{u}, \nabla f \rangle_{\mathrm{Sa}} \, dq = \langle u, \int_{\pi^{-1}(x)} f \, dq \rangle_{T_x \mathcal{M}}.$$

*where we denote by $\tilde{u}$ the unique element of $H_{(x,v)}$ such that $\pi_* \tilde{u} = u$ (the horizontal lift).*

*Proof.* The second is an immediate consequence of (18, Proposition 3.14). For the first, by definition,

$$u(f) = \langle u, \nabla f \rangle_{\mathrm{Sa}} = \langle K^{\perp} u, K^{\perp} \nabla f \rangle_{N_x \mathcal{M}}.$$

Now, $u(f) = \sum_i u_i \frac{\partial f}{\partial v_i}$ and the identification is immediate.

$\square$

**Lemma B.2.** $\tilde{\Lambda}_{\mu,r}$ *has discrete spectrum with finite multiplicity eigenvalues, and decomposes as*

$$\tilde{\Lambda}_{\mu,r} = \Delta_{H,p} + \frac{1}{r^2} \Delta_{V,q}.$$

*Proof.* It follows from (20, Section 2.1) that

$$\tilde{\Lambda}_{\mu,r} = \sigma_{1/r}^* \tilde{\Delta}_{\mu,r} \sigma_r^* = \Delta_H + \frac{1}{r^2} \Delta_V + \sigma_{1/r}^* \langle \nabla \log d(\sigma_{r*} \mu), \nabla \cdot \rangle_{\mathrm{Sa}} \sigma_r^*.$$

We first verify that

$$\sigma_{1/r}^* \langle \nabla \log d(\sigma_{r*} \mu), \nabla \cdot \rangle_{\mathrm{Sa}} \sigma_r^* = \langle \nabla \log p, \pi_* \nabla \cdot \rangle_{T\mathcal{M}} + \frac{1}{r^2} \langle \nabla \log q, K^{\perp} \nabla \cdot \rangle_{N\mathcal{M}}.$$

A change of variables yields $d(\sigma_{r*} \mu) = \sigma_r^*(d\mu) r^{D-d} \propto (\sigma_r^* p)(\sigma_r^* q)$, hence

$$\langle \nabla \log d(\sigma_{r*} \mu), \nabla \cdot \rangle_{\mathrm{Sa}} = \langle \nabla \log \sigma_r^* p, \pi_* \nabla \cdot \rangle_{T\mathcal{M}} + \langle \nabla \log \sigma_r^* q, K^{\perp} \nabla \cdot \rangle_{N\mathcal{M}},$$

where the final equality follows from $p, q$ being horizontal/vertical respectively, and we identify them with the corresponding functions on $\mathcal{M}$, $B(0,1)$. Noting that differentiation commutes with the pullback, we see

$$\begin{aligned} \langle \nabla \log d(\sigma_{r*} \mu), \nabla \cdot \rangle_{\mathrm{Sa}} \sigma_r^* &= \langle \nabla \log \sigma_r^* p, \pi_* \nabla \sigma_r^* \cdot \rangle_{T\mathcal{M}} + \langle \nabla \log \sigma_r^* q, K^{\perp} \nabla \sigma_r^* \cdot \rangle_{N\mathcal{M}} \\ &= \langle \sigma_r^* \nabla \log p, \pi_* \sigma_r^* \nabla \cdot \rangle_{T\mathcal{M}} + \langle \sigma_r^* \nabla \log q, K^{\perp} \sigma_r^* \nabla \cdot \rangle_{N\mathcal{M}} \\ &= \langle \nabla \log p, \pi_* \nabla \cdot \rangle_{T\mathcal{M}} + \frac{1}{r^2} \langle \nabla \log q, K^{\perp} \nabla \cdot \rangle_{N\mathcal{M}}, \end{aligned}$$

where the last inequality follows from the Sasaki metric being the canonical variation relative to $\pi$ (18). The final composition with $\sigma_{1/r}^*$ maps the above inner-product (function) from $T_r(\mathcal{M})$ to $T_1(\mathcal{M})$ as desired.

We now verify that
$$\Delta_{V,q} := \Delta_V + \langle \nabla \log q, K^{\perp} \nabla \cdot \rangle, \quad \Delta_{H,p} := \Delta_H + \langle \nabla \log p, \pi_* \nabla \cdot \rangle$$

have the desired properties. From (18, Definition 1.2), we have $\Delta_V(f)(x,v) = \Delta_{B(0,1)}(f|_{\pi_x^{-1}})(v)$, hence by Lemma B.1,

$$\Delta_{V,q}(f)(x,v) = \Delta_{B(0,1)}(f|_{\pi_x^{-1}})(v) + \langle \nabla \log q, \nabla f|_{\pi^{-1}(x)}(v) \rangle,$$

i.e. the weighted Laplacian on the ball. For the horizontal operator, the proposition similarly follows from Proposition B.1 as $\int_{\pi^{-1}(x)} \Delta_H f \, dq = \Delta_{\mathcal{M}} \left( \int_{\pi^{-1}(x)} f \, dq \right)$ by (18, 3.9).

$\square$

**Lemma B.3.** *The operators $\tilde{\Lambda}_{\mu,r}$, $\Delta_{V,q}$, and $\Delta_{H,p}$ are non-negative and self-adjoint.*

*Proof.* Integration by parts combined with the Neumann boundary condition yields

$$\int g\Delta_{V,q}f d\mu = \int \langle K^{\perp}\nabla g, K^{\perp}\nabla f\rangle d\mu = \int f\Delta_{V,q}g d\mu$$

$$\int f\Delta_{H,p}g d\mu = \int \langle \pi_*\nabla g, \pi_*\nabla f\rangle d\mu = \int f\Delta_{H,p}g d\mu.$$

$\square$

**Lemma B.4.** *$\tilde{\Lambda}_{\mu,r}$, $\Delta_{H,p}$, and $\Delta_{V,q}$ commute pairwise, thus there is a common orthonormal basis $\{\phi_i\}_{i=1}^{\infty}$ $L^2(T_1(\mathcal{M}, \mu_{\mathrm{Sa}})$ of smooth eigenfunctions such that*

$$\Delta_{H,p}\phi_i = \lambda_i^H \phi_i, \quad \Delta_{V,q}\phi_i = \lambda_i^V \phi_i, \quad \tilde{\Lambda}_{\mu,r}\phi_i = (\lambda_i^H + \frac{1}{r^2}\lambda_i^V)\phi_i =: \tilde{\lambda}_{i,r}\phi_i. \tag{1}$$

*Note that these eigenfunctions and spectra $\lambda_i^H, \lambda_i^V$ are invariant for $0 < r < r_{\mathcal{M}}$*

*Proof.* $\langle \nabla \log \sigma_r^* p, \pi_*\nabla\cdot\rangle_{T\mathcal{M}}, \langle \nabla \log \sigma_r^* q, K^{\perp}\nabla\cdot\rangle_{N\mathcal{M}}$ are operators on the horizontal and vertical tangent spaces, respectively, hence they commute, and they commute with $\Delta_V, \Delta_H$ respectively as seen in the argument of (18, Proposition 1.6). This implies that the operators can be simultaneously diagonalized, i.e., the existence of a common basis of eigenfunctions. $\square$

**Lemma B.5.** *For $\lambda_i(\tilde{\Lambda}_{\mu,r}) \leq \lambda_1^V/r^2$, $\phi_i(\tilde{\Lambda}_{\mu,r}) = \phi_i(\Delta_p) \circ \pi$ and $\lambda_i(\tilde{\Lambda}_{\mu,r}) = \lambda_i(\Delta_p)$.*

*Proof.* By Lemma B.1, if $\phi_i$ is an eigenfunction of $\Delta_{V,q}$, its fiber-wise restriction $\phi_i|_{\pi^{-1}(x)}$ must be an eigenfunction of the corresponding weighted Laplacian on the ball. As its eigenvalue is smaller than $\lambda_1^V/r^2$, it must correspond to the $0$ eigenvalue. This implies $\phi_i$ is constant fibre-wise, in particular $\phi_i(x,v) = \int_{\pi^{-1}(x)} \phi_i|_{\pi^{-1}(x)} dq$. Thus, by Proposition B.1,

$$\tilde{\Lambda}_{\mu,r}\phi_i = \Delta_{H,p}\phi_i + \Delta_{V,q}\phi_i = \Delta_{H,p}\phi_i = \Delta_{\mathcal{M},p}\int_{\pi^{-1}(x)} \phi_i|_{\pi^{-1}(x)} dq.$$

Thus for $\phi_i$ to be an eigenfunction, it must be that $\int_{\pi^{-1}(x)} \phi_i|_{\pi^{-1}(x)} dq$, as a function on $\mathcal{M}$, is an eigenfunction of $\Delta_{\mathcal{M},p}$, call it $f$. Further, for all $v$, $\phi_i(x,v) = f(x)$, or in other words, $\phi = f \circ \pi$ as desired, and $\tilde{\Lambda}_{\mu,r}\phi_i = \Delta_{\mathcal{M},p}f \circ \pi = \lambda_i^H f \circ \pi$. Conversely, any $f$ an eigenfunction of $\Delta_{\mathcal{M},p}$ with eigenvalue $\lambda < \lambda_1^V/r^2$ corresponds to some $\phi_i$ as above. $\square$

## B.2. Perturbation Argument

Our first aim is to analyze the effects of the noise on the (weighted) manifold Laplacian, namely to quantify the error between $\Delta_{T_r(\mathcal{M})}$ representing the Laplacian operator of the "noisy" manifold $T_r(\mathcal{M})$, and the "noiseless" manifold Laplacian $\Delta_{\mathcal{M}}$, particularly as it pertains to the spectral data associated to the principal eigenvalues. More precisely, we will be concerned with eigenvalues below a threshold of the order $r^{-2}$, showing that Proposition 2.1 holds approximately for $\Delta_{T_r(\mathcal{M})}$. We study both the unweighted case, where the sampling density $\mu$ on $T_r(\mathcal{M})$ is uniform w.r.t. the induced measure from $\mathbb{R}^D$, and the weighted case where smooth, non-degenerate $\mu$ factors along horizontal and vertical spaces. Throughout this section the assumptions are the same as in Section 2.

The main idea is that the Sasaki metric closely approximates the induced metric on $T_r(\mathcal{M})$. It follows from (20, Proposition 6) that, under rescaling, the residual between the induced metric $\langle\cdot,\cdot\rangle_r$ on $T_r(\mathcal{M})$ and the Sasaki metric $\langle\cdot,\cdot\rangle_{\mathrm{Sa},r}$ is on the order of the tube width.

**Lemma B.6** (Sasaki metric perturbation). *For the rescaled induced metric $\langle\cdot,\cdot\rangle_r$, let $A_{r,(x,v)}$ denote its relative distortion w.r.t the rescaled Sasaki metric $\langle\cdot,\cdot\rangle_{\mathrm{Sa},r} := \langle\sigma_r^*\cdot, \sigma_r^*\cdot\rangle_{\mathrm{Sa}}$. Then $\|I - A_r\|_{\infty} \leq Cr$ for a constant $C$ depending only on the manifold $\mathcal{M}$.*

*Proof.* From (20, Proposition 6), we have

$$\langle \eta, \zeta \rangle_r = \langle \eta, \zeta \rangle_{\mathrm{Sa},r} - \frac{1}{3} \langle \eta^\perp, R_W^* \zeta^\perp \rangle_{\mathrm{Sa},x} + r \mathbf{r}_r(\eta, \zeta)$$

for some $\mathbf{r}_r$ smooth and uniformly bounded for all $r \leq 1$, and $R_W$ the Weingarten map (or shape operator) (17, Chapter 3). The middle term, a function of the curvature of the ambient manifold, vanishes as the ambient Euclidean space is flat. Hence we get a perturbation of the desired order. $\square$

**Corollary B.7.** *Let $\mu$ be the induced measure and $\mu_{\mathrm{Sa}}$ the measure induced by the Sasaki metric. There exists a constant $C$ depending on $\mathcal{M}$ such that*

$$\|\mathbf{1} - d\mu/d\mu_{\mathrm{Sa}}\|_\infty \leq (Cr+1)^d - 1 = O(Cr), \quad \|I - A_r\|_\infty \leq Cr.$$

*Proof.* From Proposition B.6 we have
$$\|I - A_r\|_\infty \leq r\|\mathbf{r}_r\|_\infty \leq Cr,$$
for some $C$ depending on $\mathcal{M}$. Now, $d\mu/d\mu_{\mathrm{Sa}} = \det A_r$, and the above bound on the operator norm implies
$$(1 - Cr)^d \leq \det A_r \leq (Cr+1)^d,$$

from which the inequality follows. $\square$

Next, to compare the spectral decompositions of $\tilde{\Delta}_{\mu,r}$ and $\Delta$, we use a variational argument, studying the Dirichlet forms of the two operators. As it will be of interest to us to compare operators relative to different underlying densities (Sasaki vs induced), this requires an identification between the corresponding spaces of Sobolev functions $\mathcal{H}^2(\mu), \mathcal{H}^2(\mu_{\mathrm{Sa}})$. Even though these spaces coincide, as a simple analysis such as (33, Proposition 5.2.2) reveals, some care is needed as the norms are not the same, i.e. the $L^2$ norm of the same function will differ depending on the measure. What we require is that they are *nearly* isometric, that these norms can be made arbitrarily close as $r \to 0$. For operators on different domains, in this case the two sobolev spaces, one must find an identification between their domains for which the identified operators are comparable. This leads to the technical notion of *quasi-unitary* equivalence.

Let $\Delta, \Delta'$ be non-negative linear operators on $\mathcal{H}, \mathcal{H}'$ respectively. $\Delta, \Delta'$ are $\delta$-quasi-unitarily equivalent if there exists $J : \mathcal{H} \to \mathcal{H}'$, $J' : \mathcal{H}' \to \mathcal{H}$ such that

$$\|J\| \leq 2, \quad \|J^* - J'\| \leq \delta,$$
$$\|(I - J'J)(\Delta + I)^{-1}\| \leq \delta, \quad \|(I - JJ')(\Delta' + I)^{-1}\| \leq \delta,$$
$$\|J(\Delta + I)^{-1} - (\Delta' + I)^{-1}J\| \leq \delta.$$

From this framework, the main result we will need is (33, Theorem 5.2.6), which we restate as follows.

**Proposition B.8.** *Let $\Delta_\mu, \Delta'_\nu$ be weighted Laplacians with relative distortion $A$ and $\rho := d\nu/d\mu$. Let $J : H^1(\mu) \to H^1(\nu)$ and $J' : H^1(\nu) \to H^1(\mu)$ be the trivial identification maps $Ju = u, J'f = f$. Then for*

$$\delta := \max\{\|\rho^{1/2} - \rho^{-1/2}\|_\infty, \|\rho^{-1/2}A^{1/2} - \rho^{1/2}A^{-1/2}\|_\infty\},$$
$$\hat{\delta} := \max\{\|\mathbf{1} - \rho\|_\infty, \|I - A\|_\infty\},$$

*$\Delta_\mu, \Delta'_\nu$ are $4\delta$-quasi-unitarily equivalent. If $\hat{\delta} \leq 1/2$ then they are also $16\hat{\delta}$-quasi-unitarily equivalent.*

The following is a consequence of (33, Proposition 4.3.1) and the previous results, providing a general relationship between the Sasaki and induced spectra.

**Corollary B.9.** *There exists a constant $C$ depending on $\mathcal{M}$ such that, for $\delta(r) := (Cr+1)^d - 1 = O(r)$, if $\delta(r) \leq 1/2$,*

$$|\lambda_k(\tilde{\Lambda}_r) - \lambda_k(\Lambda_r)| \leq 8\lambda_k(\tilde{\Lambda}_r)\delta(r).$$

*Further, there exists a constant $C_\lambda$ such that, for $\delta(r) \leq (1 + \lambda + C_\lambda)^{-1}$,*

$$\|P_\lambda(\tilde{\Lambda}_r) - P_\lambda(\Lambda_r)\| \leq 16C_\lambda\delta(r),$$

*in particular, for any $\lambda_k(\tilde{\Delta}_r) \leq \lambda$ simple, $\phi_k(\tilde{\Lambda}_r), \phi_k(\Lambda_r)$ unit, we have*

$$\|\phi_k(\tilde{\Lambda}_r) - \phi_k(\Lambda_r)\| \leq [2C_\lambda + 3(\lambda_k(\tilde{\Delta}_r) + 1)]\delta(r).$$

**Corollary B.10.** *There exists a constant $C$ depending on $\mathcal{M}$ such that, if $Cr \leq 1/2$,*

$$|\lambda_k(\tilde{\Lambda}_{\mu,r}) - \lambda_k(\Lambda_{\mu,r})| \leq 8\lambda_k(\tilde{\Lambda}_{\mu,r})Cr.$$

*Further, there exists a constant $C_\lambda$ such that, for $Cr \leq (1 + \lambda + C_\lambda)^{-1}$,*

$$\|P_\lambda(\tilde{\Lambda}_r) - P_\lambda(\Lambda_r)\| \leq 16C_\lambda Cr,$$

*in particular, for any $\lambda_k(\tilde{\Delta}_r) \leq \lambda$ simple, $\phi_k(\tilde{\Lambda}_{\mu,r}), \phi_k(\Lambda_{\mu,r})$ unit, we have*

$$\|\phi_k(\tilde{\Lambda}_{\mu,r}) - \phi_k(\Lambda_{\mu,r})\| \leq [2C_\lambda + 3(\lambda_k(\tilde{\Delta}_r) + 1)]Cr.$$

*Proof of Theorem 3.1.* This result follows from the above results in tandem with Proposition 2.1, part 5.

$\square$

*Proof of Corollary 3.2.* This corollary follows immediately due to the relationship $\mathcal{P}(\mu) = \lambda_1(\Delta_\mu)^{-1}$. $\square$

# C. Multi-Scale Spectral Growth

## C.1. Packings

A $\varepsilon - \|\cdot\|$ packing of a space $\mathcal{X}$, $\mathcal{X}_\varepsilon = \{x_1^\varepsilon, \ldots, x_{\mathrm{Pack}(\mathcal{X},\varepsilon,\|\cdot\|)}^\varepsilon\}$, is a maximal collection of points $x_i^\varepsilon$ such that $\min \|x_i^\varepsilon - x_j^\varepsilon\| \geq \varepsilon$. The packing number $\mathrm{Pack}(\mathcal{X}, \varepsilon, \|\cdot\|)$ is exactly the quantity of elements in the packing. Covering numbers $\mathrm{Cover}(\mathcal{X}, 2\varepsilon, \|\cdot\|)$ are similarly defined, but rather than a separation requirement, the union of the $\varepsilon$ neighborhoods of the cover elements should be the whole space $\mathcal{X}$, and the cover should be a minimal collection that achieves this. In both notations, we suppress the dependence on the norm when it is understood.

It follows that $\mathcal{X}_\varepsilon$ is also a $2\varepsilon$ cover of $\mathcal{X}_\varepsilon$, $\mathrm{Pack}(\mathcal{X}, \varepsilon) \geq \mathrm{Cover}(\mathcal{X}, \varepsilon)$ as otherwise, if a point was $2\varepsilon$ distance from every member of this set, it could be included while maintaining the $\varepsilon$ separation property, contradicting the maximality of this set.

## C.2. Proof of Theorem 3.3

We prove Theorem 3.3 in two steps. First, we focus on upper-bounding the counting function. We follow the details of (26).

**Lemma C.1.** *Let $\mathcal{X}$ be a bounded domain with smooth boundary. Then,*

$$N(C(4,\varepsilon)\varepsilon^{-2}). \leq \mathrm{Pack}(\mathcal{X}, \varepsilon)/(4C^2(4, \varepsilon)).$$

*Proof.* We aim to construct a collection of orthogonal Neumann functions with controlled dirichlet form growth. By the min-max characterization, this upper-bounds the growth of corresponding eigenvalues. The simplest way to do this is to ensure that our functions have no common support, hence $L^2$ orthogonality follows trivially. This is naturally achieved by a packing, although the boundary of our domain requires a slightly specialized approach. That is, near the boundary, the ball around a point in the subspace topology may be very degenerate, necessitating a more careful construction. This task has already been undertaken in (26), and we reference the essential results.

Let $\varepsilon, N > 0$. By (26, Corollary 2.3), if $\sup_{x \in \mathcal{X}} 4C^2(4,\varepsilon)\mu(B(x,\varepsilon)) \leq \mu(\mathcal{X})/N$ then one can construct $A_1, \ldots, A_N \subseteq \mathcal{X}$ such that

$$\mu(A_i) \geq \frac{\mu(\mathcal{X})}{2C(4,\varepsilon)}, \quad d(A_i, A_j) \geq 3\varepsilon.$$

Let $A_i^\varepsilon$ denote the $\varepsilon-$fattening of $A_i$, and let $I_\varepsilon$ be the collection of indices such that $\mu(A_i^\varepsilon) \geq \mu(A_i)/(N/2)$. By subadditivity, it is immediate that $|I_\varepsilon| \leq N/2$. Hence, for any $i \in I_\varepsilon^c$, by (26, Proposition 3.1), one can select $f_i$ unit norm with Neumann boundary conditions supported on $A_i^\varepsilon$ such that $R(f_i) \leq 4C(4,\varepsilon)/\varepsilon^2$. As the support of these functions is disjoint, they are an orthonormal set of at least $N/2$ functions, hence

$$\lambda_{N/2} \leq \frac{4C(4,\varepsilon)}{\varepsilon^2}.$$

It follows that $4C^2(4,\varepsilon)\sup_{x\in\mathcal{X}}\mu(B(x,\varepsilon)) \leq \mu(\mathcal{X})/N$ implies $\lambda_{N/2} \leq 4C(4,\varepsilon)/\varepsilon^2$. Notice that

$$\text{Pack}(\mathcal{X},\varepsilon) \leq \mu(\mathcal{X})/\inf_{x\in\mathcal{X}}\mu(B(x,\varepsilon))$$

hence $4C^2(4,\varepsilon)\sup_{x\in\mathcal{X}}\mu(B(x,\varepsilon)) \leq \mu(\mathcal{X})/N$ is satisfied for $4C^2(4,\varepsilon)N \leq \text{Pack}(\mathcal{X},\varepsilon)\frac{\inf_{x\in\mathcal{X}}\mu(B(x,\varepsilon))}{\sup_{x\in\mathcal{X}}\mu(B(x,\varepsilon))}$, that is

$$\lambda_{\text{Pack}(\mathcal{X},\varepsilon)/(4C^2(4,\varepsilon))\frac{\inf_{x\in\mathcal{X}}\mu(B(x,\varepsilon))}{\sup_{x\in\mathcal{X}}\mu(B(x,\varepsilon))}} \leq \frac{4C(4,\varepsilon)}{\varepsilon^2},$$

as desired. $\qquad\square$

For the remaining inequality, we follow (27).

**Lemma C.2.** *For $\mathcal{X}$ compact and convex or closed without boundary, there exists a constant $C'$ depending on $\mathcal{X}$ such that*

$$\text{Pack}(\mathcal{X},\varepsilon) \leq N(C'\varepsilon^{-2}/C(2,\varepsilon)).$$

*Proof.* The structure of this proof is similar in flavor to Lemma C.1. We again subdivide our domain, this time via packing, and consider the moments of different Laplacian eigenfunctions restricted to these disjoint balls. Precisely, for $\mathcal{X}_\varepsilon$, consider the map

$$b(f) = \begin{bmatrix} \frac{1}{\mu(B(x_1^\varepsilon,\varepsilon))}\int_{B(x_1^\varepsilon,\varepsilon)}f d\mu \\ \vdots \\ \frac{1}{\mu(B(x_{\text{Pack}(\mathcal{X},\varepsilon)}^\varepsilon,\varepsilon))}\int_{B(x_{\text{Pack}(\mathcal{X},\varepsilon)}^\varepsilon,\varepsilon)}f d\mu \end{bmatrix} \in \mathbb{R}^{\text{Pack}(\mathcal{X},\varepsilon)}.$$

Let $\{\phi_i\}_{i=1}^\infty$ be unit Laplacian eigenfunctions, and $E(\lambda) = \text{span}\{\phi_i : \lambda_i \leq \lambda\}$ for some $\lambda > 0$. Suppose that there exists $f \in E(\lambda)$ such that $b(f) = 0$. To fall in the null space of $b$, $f$ must be mean 0 across each of the given subdomains. Intuitively, this requires strong variation for our function, which we quantify via Poincaré-type inequalities on each of the balls, forcing the corresponding Laplacian eigenvalue to be large.

Making this precise, we have $\frac{1}{\mu(B(x_i^\varepsilon,\varepsilon))}\int_{B(x_i^\varepsilon,\varepsilon)}f d\mu = 0$ for all $i \in \text{Pack}(\mathcal{X},\varepsilon)$. In particular, we can use (27, Proposition 2.1, Proposition 2.1), and the realization of $\mathcal{X}_\varepsilon$ as a $2\varepsilon$ cover to bound

$$\int f^2 d\mu < \sum_{i=1}^n \int_{B(x_i^\varepsilon,2\varepsilon)} f^2 d\mu \leq C'\varepsilon^2 \sum_{i=1}^n \int_{B(x_i^\varepsilon,2\varepsilon)} \|\nabla f\|^2 d\mu \leq C'C(2,\varepsilon)\varepsilon^2 \int_{\mathcal{X}} \|\nabla f\|^2 d\mu,$$

where $C'$ is a constant that depends on the dimension. Note that the first inequality is strict as if no two balls had non-trivial overlap then another ball could be adjoined to the packing. To see the final inequality, we must show that every $x \in \mathcal{X}$ is contained in at most $C(2,\varepsilon)$ of the $B(x_i^\varepsilon,2\varepsilon)$ balls. Take $x \in B(x_i^\varepsilon,2\varepsilon)$. Observe that $x_i^\varepsilon \in B(x,2\varepsilon)$, and in particular, for any two such points covering $x$, $x_i^\varepsilon, x_j^\varepsilon \in B(x,2\varepsilon)$. Thus, by the separation condition, there can be at most $\text{Pack}(B(x,2\varepsilon),\varepsilon)$ $x_i^\varepsilon$ covering $x$.

For $f \in E(\lambda)$ we have

$$\int_{\mathcal{X}} f^2 d\mu \geq \lambda^{-1} \int_{\mathcal{X}} \|\nabla f\|^2 d\mu,$$

hence $b(f) = 0$ implies $\lambda^{-1} < C'C(2,\varepsilon)\varepsilon^2$. Thus for $\lambda \geq C'C(2,\varepsilon)\varepsilon^2$, $b$ is an injection. In particular, this implies that $\dim E(\varepsilon^{-2}/(C'C(2,\varepsilon)) \leq \text{Pack}(\mathcal{X},\varepsilon)$, that is

$$\lambda_{\text{Pack}(\mathcal{X},\varepsilon)} \geq \varepsilon^{-2}/(C'C(2,\varepsilon)).$$

$\qquad\square$

# D. Further Data Details

## D.1. Synthetic Data

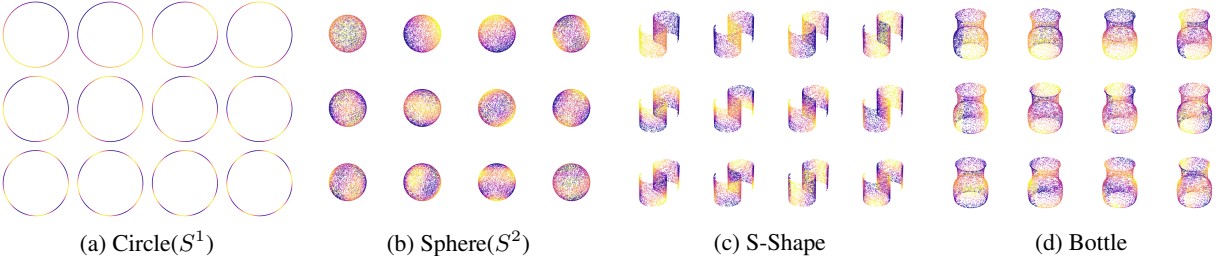

(a) Circle($S^1$)  (b) Sphere($S^2$)  (c) S-Shape  (d) Bottle

Figure 7: Samples from the four synthetic manifolds used in our experiments colored by the first twelve $\phi_i(\Delta_\mathcal{M})$ estimated using Diffusion Maps. In subfigure d), only the first 3 ambient coordinates of the Bottle manifold are shown. The manifolds have intrinsic dimensions $d_{S^1} = 1$, $d_{S^2} = 2$, $d_{SShape} = 2$, $d_{Bottle} = 2$.

Throughout our experiments we will use four synthetic smooth manifolds: The unit circle $S^1$, the unit sphere $S^2$, the S-Shape, and a new manifold that we created which we will refer to as the Bottle(See Figure 7). This latter manifold is parametrized by:

$$f(x_1, x_2) = \begin{bmatrix} \sin(\pi x_2)(1 + \sigma(\sin(\pi x_1))) \\ \cos(\pi x_2)(1 + \sigma(\sin(\pi x_1))) \\ sign(\pi x_1)(\cos(\pi x_1) - 1) \\ 2x_2 \end{bmatrix}$$

|  | $d$ | $D_\mathcal{M}$ |
|---|---|---|
| $S^1$ | 1 | 2 |
| $S^2$ | 2 | 3 |
| S-Shape | 2 | 3 |
| Bottle | 2 | 4 |

Table 1: Synthetic manifolds' dimension

for $(x_1, x_2) \in [0, 1] \times [0, 1]$ and $\sigma$ the sigmoid function.

The intrinsic dimension $d$ and 'usual' ambient dimension $D_\mathcal{M}$(before they are embedded again in an ambient space $\mathbb{R}^D$ with $D_\mathcal{M} \leq D$ where noise will be added) of the synthetic manifolds can be found in Table 1. We sample points uniformly from $S^1$ and $S^2$, while from the S-Shape and the Bottle we sample coordinates uniformly and map them to the 'usual' ambient space. This sample is then embedded in a larger dimensional ambient space of dimension $D$ where noise is added uniformly on a ball of radius $r$ in the normal space which has dimension $d_n = D - d$. For our four synthetic manifolds we analytically compute the normal space at every point. We repeat this process for various $r \in [0, \tau)$ and $d_n$. We sample as many points as we consider necessary depending on the experiment, but for most cases where we are not interested in exploring the effects of sample size, we sample $n = 5000$ points.

## D.2. Real Data

For our real data experiments we use three Molecular Dynamic Simulations(MDS) datasets: Toluene(Tol), Malonaldehyde(Mal), and Ethanol(Eth). MDS simulations dynamically generate atomic configurations which, due to interatomic interactions, exhibit non-linear, multiscale, non-i.i.d. noise, as well as non-trivial topology and geometry. MDS are a heavily used tool, with many hours of high power computing devoted to them. For more information on the MDS dataset we refer the reader to (28).

|  | $d$ | $D_\mathcal{M}$ |
|---|---|---|
| Tol | 1 | 50 |
| Mal | 2 | 50 |
| Eth | 2 | 50 |

Table 2: Real manifolds' dimension

MDS data is originally generated in $\mathbb{R}^{3 \cdot N_{atoms}}$ coordinates. We process this raw data to ensure the invariance to translation and rotation before we estimate the Laplace operators. For this, we obtain an Euclidean group-invariant featurization of the atomic coordinates as a vector of planar angles $a_i \in \mathbb{R}^{3 \cdot \binom{N_a}{3}}$ representing the planar angles formed by triplets of atoms in the molecule. We then perform SVD on this featurization, and project the data onto the top $D = 50$ singular vectors to remove linear redundancies. Thus, all three MDS datasets will have the same ambient dimension $D_\mathcal{M} = 50$. Their intrinsic dimensions $d$ can be found in Table 2 while visualizations are provided in Figure 8.

We add noise uniformly distributed on a ball of radius $r$ in the ambient space for various $r$ and $d_n$. Noise is added in the ambient space because, as opposed to the synthetic data, we don't have access to the reach $\tau$ or to the normal bundle of the MDS manifold. We do this for various $r$. We sample $n = 7500$ points for all our experiments.

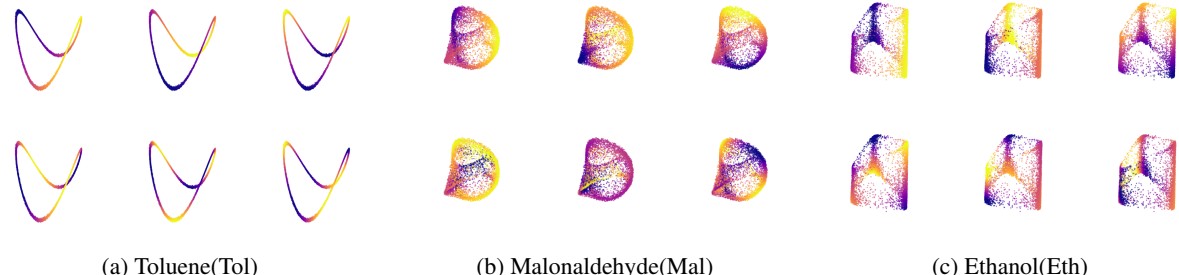

| (a) Toluene(Tol) | (b) Malonaldehyde(Mal) | (c) Ethanol(Eth) |

Figure 8: Samples from the three MDS datasets: Toluene(Tol), Malonaldehyde(Mal), and Ethanol(Eth) colored by the first twelve $\phi_i(\Delta_\mathcal{M})$ estimated using Diffusion Maps. The manifolds are embedded in $\mathbb{R}^3$ using $\phi_{1,2,3}(\Delta_\mathcal{M})$, but they have intrinsic dimensions $d_{Tol} = 1, d_{Mal} = 2, d_{Eth} = 2$.

## E. Further Details on Validation Experiments

We conduct a series of experiments validating the threshold phenomenon and the robustness of low-frequency eigenpairs to noise predicted by our theoretical results on the datasets described in Appendix D. For this purpose, we need to estimate various Laplace operators and their eigenpairs as they have no analytical formulas that we can use($S^d$ being the exception). We generate multiple datasets representing samples from $T_r(\mathcal{M})$ for multiple increasing values of the two main noise parameters of interest: $r$(the radius of the tubular neighborhood) and $d_n$(the dimension of the normal space which is related to the ambient dimension by $D = d_n + d$). Theorem 3.1 predicts that low-frequency eigenvectors $\phi_i(\Delta_{T_r(\mathcal{M})})$ are stable w.r.t. to these, while the noise threshold $\lambda_1^V / r^2$ acts as a delimiter between the noiseless and noisy eigenvectors. As such, we develop both quantitative and qualitative methods to investigate this behavior.

### E.1. Estimating the Laplace Operators and their Eigenpairs

We rely on Diffusion Maps(1) to estimate the Laplace operators $\Delta_\mathcal{M}, \Delta_{T_r(\mathcal{M})}$, and their spectral decompositions. We select the bandwidth $\epsilon$ required to compute the affinity matrix on a per dataset basis using the geometric self-consistency algorithm of (41). We do so for all manifolds described in Section D, for many tubular neighborhood radii $r \in (0, \tau)$(when the reach is known), and dimensions of the normal space $d_n = D - d$. For most of our experiments we use an embedding size of $m = 24$, but in some case we use $m > 24$.

### E.2. Qualitative Verification of the Theory

To validate the threshold phenomenon and the robustness of low-frequency eigenpairs to noise predicted by Theorem 3.1 we observe the behavior of the eigenvalues $\lambda_i(\Delta_{T_r(\mathcal{M})})$ and eigenvectors $\phi_i(\Delta_{T_r(\mathcal{M})})$ of the Laplacians estimated in Section E.1 as $r$ and $d_n$ are varied.

First, in Figures 9 and 10 we focus on the paths which the eigenvalues trace as $r$ increases. Furthermore, we keep track of the noise threshold and observe the effect that it has on the stability of the $\lambda_i(\Delta_{T_r(\mathcal{M})})$. We notice that, as the theory predicts, the eigenvalues well under $\lambda_1^V / r^2$ are almost constant, while the higher frequency ones become volatile with a downward trend as they approach or surpass the noise threshold. Furthermore, we observe that increasing the noise dimension $d_n$(and implicitly increasing $D$) has no bearing on this effect.

Second, in Figures 11 and 12 we perform a similar experiment, but observe the behavior of the eigenvectors $\phi_i(\Delta_{T_r(\mathcal{M})})$ instead. In order to test their degree of corruption by noise, we color the coordinate spaces of the manifolds by $\phi_i(\Delta_{T_r(\mathcal{M})})$. Visually, there should be a clear correlation between the values of the eigenvectors and the manifold coordinates, indicating that $\phi_i(\Delta_{T_r(\mathcal{M})})$ correctly captures the structure of $\mathcal{M}$. When that is not the case, we expect to see random patterns over the coordinate space. Additionally, since for the synthetic manifolds we know the noise/normal space coordinates at each sampled point, we fit polynomial regression models that use these as inputs and the $\phi_i$ as targets. If the model is able to predict the embedding coordinates from the noise coordinates, we can safely conclude that $\phi_i(\Delta_{T_r(\mathcal{M})})$ encodes pure noise. In our experiments we use polynomials of degree 2. Since we do not know the coordinate spaces of the MDS manifolds, we use the lowest-frequency $\phi_{1,2,3}(\Delta_\mathcal{M})$ of the noiseless manifolds instead. These experiments, which confirm our predictions,

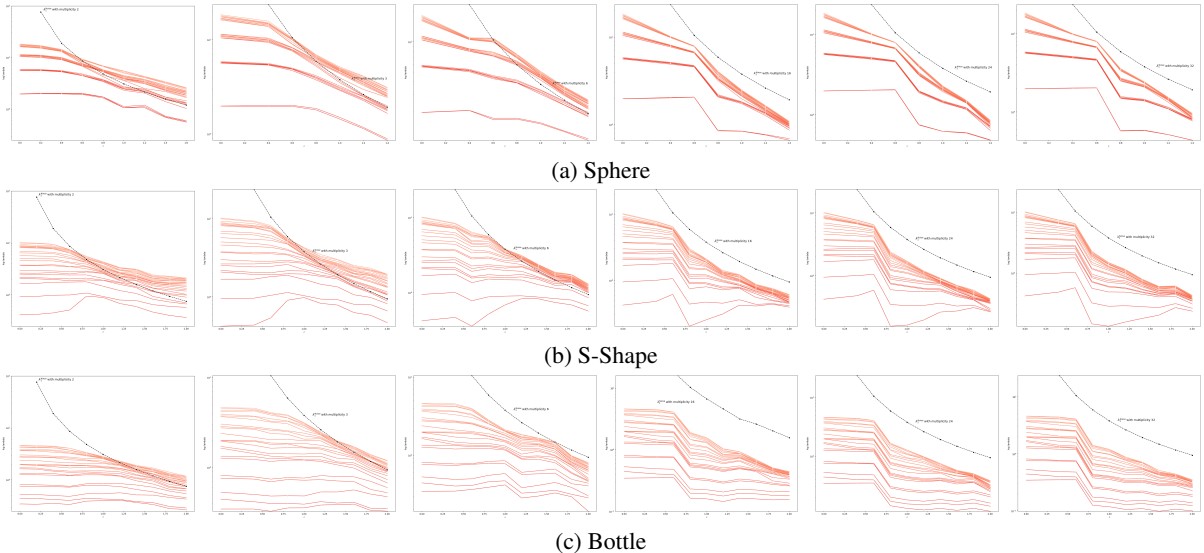

(a) Sphere

(b) S-Shape

(c) Bottle

Figure 9: Stability w.r.t to $r$ and $d_n$ of the low-frequency eigenvalues for three synthetic manifolds: Sphere(a), S-Shape(b), and Bottle(c). Each path corresponds to one $\lambda_i(\Delta_{T_r(\mathcal{M})})$ with its log value on the y-axis evolving as $r$ increases from left to right on the x-axis. The frequency of the spectrum increases in the up y-axis direction. The noise level $\lambda_1^V/r^2$ is represented by the black dotted line. On each row we have, from left to right, increasing noise dimensions $d_n \in \{2, 3, 6, 16, 24, 32\}$. Note that the eigenvalues remain mostly constant as long as they are below the noise threshold. However, this changes when they approach or surpass $\lambda_1^V/r^2$. This phenomenon is largely independent of $d_n$.

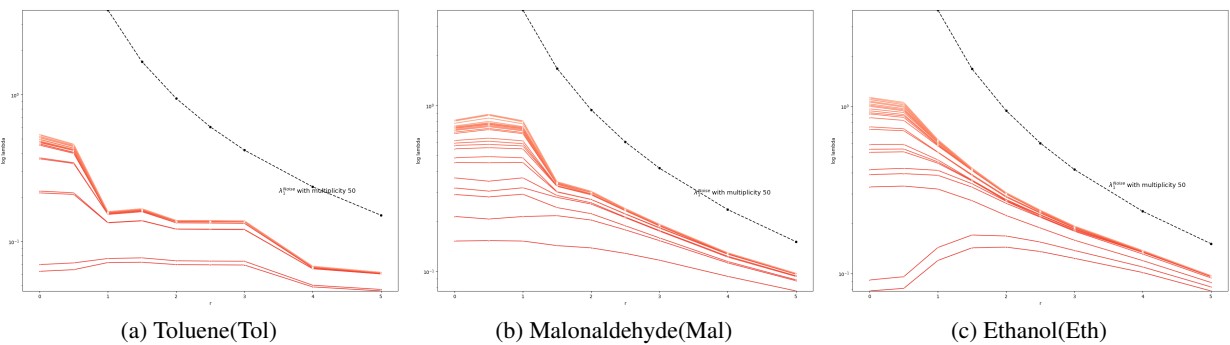

(a) Toluene(Tol)

(b) Malonaldehyde(Mal)

(c) Ethanol(Eth)

Figure 10: Stability w.r.t to $r$ of the low-frequency eigenvalues for three real MDS datasets. Each path corresponds to one $\lambda_i(\Delta_{T_r(\mathcal{M})})$ with its log value on the y-axis evolving as $r$ increases from left to right on the x-axis. The frequency of the spectrum increases in the up y-axis direction. In all plots $d_n = 50$ and, as opposed to the synthetic data, the noise is added in all ambient directions. The noise level $\lambda_1^V/r^2$ is represented by the black dotted line. Note that the eigenvalues remain mostly constant as long as they are below the noise threshold.

are summarized in Figures 11 and 12, while we display how the regression models can identify 'noise' $\phi_i(\Delta_{T_r(\mathcal{M})})$ in Figure 14.

# F. Further Details on VAE and LTSA Experiments

## F.1. VAE Experiments

VAE(Variational Autoencoder) (12) is a generative model that learns a probabilistic latent representation of data by combining deep neural networks with variational inference. It optimizes a lower bound on the data likelihood by encoding inputs into a latent space with a Gaussian prior and decoding samples back into the original space.

We repeat an adapted version of the eigenvector qualitative experiments outlined in Section E.2 and observe that VAE embeddings behave similarly to those obtained by DM. However, there is no natural way to evaluate whether a VAE's latent coordinate $\phi_i$ is low or high frequency. For this purpose we use the regression procedure outlined in Section E.2. Furthermore, we evaluate the ability of the decoder to reconstruct the output without using $\phi_i$, but using all the other coordinates. If the polynomial regression model is able to predict $\phi_i$ from noise and if the decoder doesn't require $\phi_i$ for proper reconstruction, we say that $\phi_i$ is high-frequency. Conversely, if $\phi_i$ is indispensable for reconstruction and cannot be predicted from noise, then we call this latent coordinate low-frequency. We observe that this latter set is stable w.r.t. to both $r$ and $d_n$. Furthermore, if the VAE has more capacity than need, then it will use the extra coordinates to encode noise. Our experiments are summarized in Figure .

For all our VAE embeddings we use the same network which has an encoder with FC layers of sizes (64, 128), an embedding size of $m = 4$, and a decoder with FC layers of sizes (128, 64). We use Layer Normalization (42) and GELU activation (43) between the hidden layers, a batch size of 256, Adam optimizer (44), and a weight of 0.1 of the KL-Divergence loss relative to the reconstruction loss.

## F.2. LTSA Experiments

LTSA(Local Tangent Space Alignment) (13) is a classical manifold learning algorithm that does not rely on Laplacian estimation. Instead, it computes local linear coordinates for the neighbors of each point and aligns these local representations to obtain a global embedding. We repeat the eigenvector qualitative experiments outlined in Section E.2 and observe that LTSA computed embeddings behave exactly the same as those obtained via DM and in complete agreement with Theorem 3.1.

For all our LTSA embeddings we use an embedding size of $m = 12$ and $k = 32$ nearest neighbors without a cutoff radius. We only run these experiments on the synthetic data. Our results are summarized in 14. For all our LTSA embeddings we use an embedding size of $m = 12$ and $k = 32$ nearest neighbors without a cutoff radius. We only run these experiments on the synthetic data.

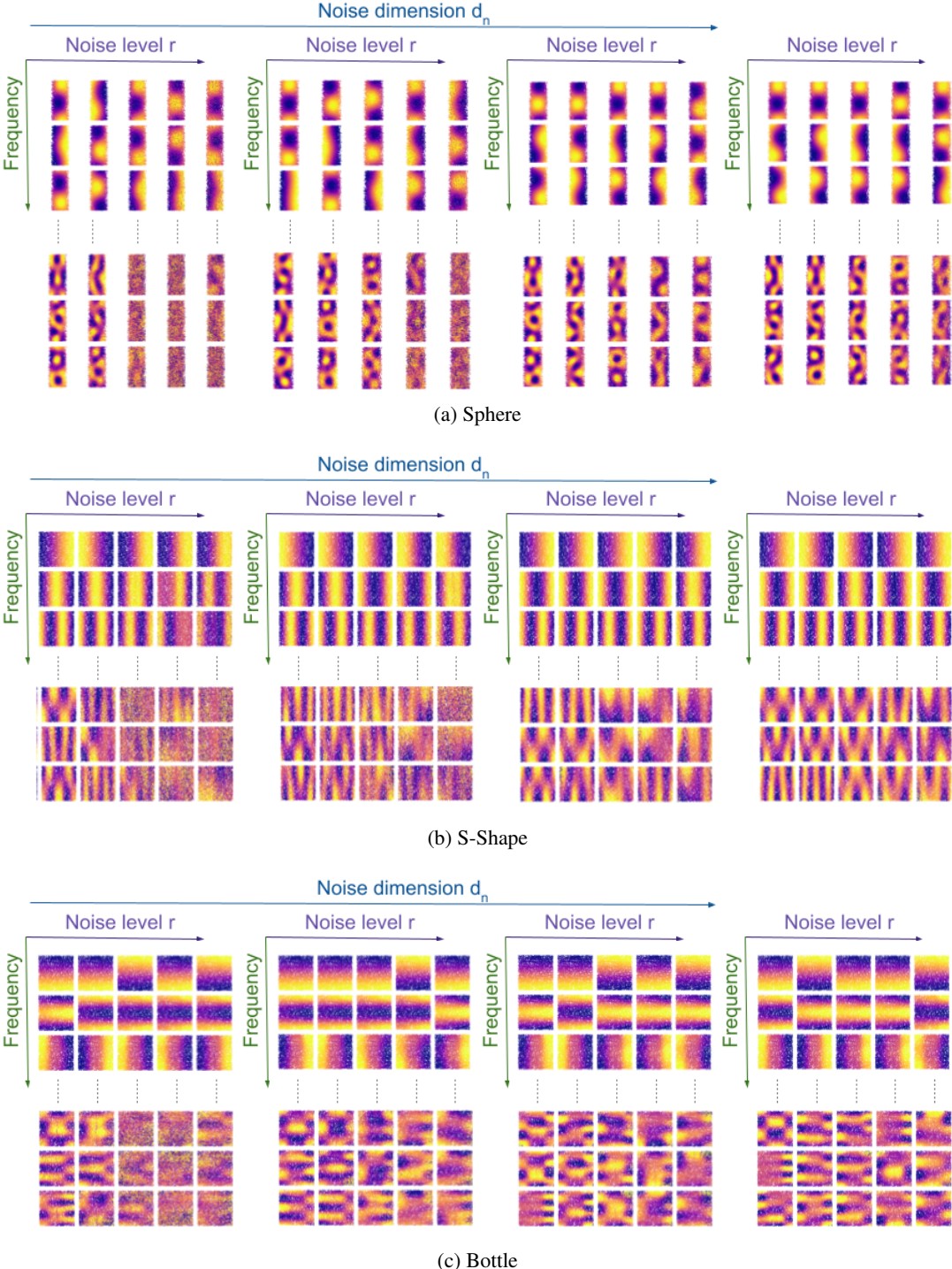

(a) Sphere

(b) S-Shape

(c) Bottle

Figure 11: Stability w.r.t to $r$ and $d_n$ of the low-frequency eigenvectors for three synthetic manifolds: Sphere(a), S-Shape(b), and Bottle(c). We color the coordinate spaces of the manifolds by low-frequency(top) and high-frequency(bottom) $\phi_i(\Delta_{T_r(\mathcal{M})})$. Notice the stability of the former as $r$ increases and the corruption of the latter once the noise threshold is passed. On each column we have, from left to right, increasing noise dimensions $d_n \in \{4, 8, 16, 24\}$. We notice, interestingly, that as $d_n$(and thus $D = d + d_n$) increases, the effect of noise on the eigenvectors with the same frequency is mitigated. This is to be expected as $\lambda_1^V$ increases with the dimension, however a more detailed perturbation analysis is necessary to rigorously verify that the perturbation magnitude decreases with dimension.

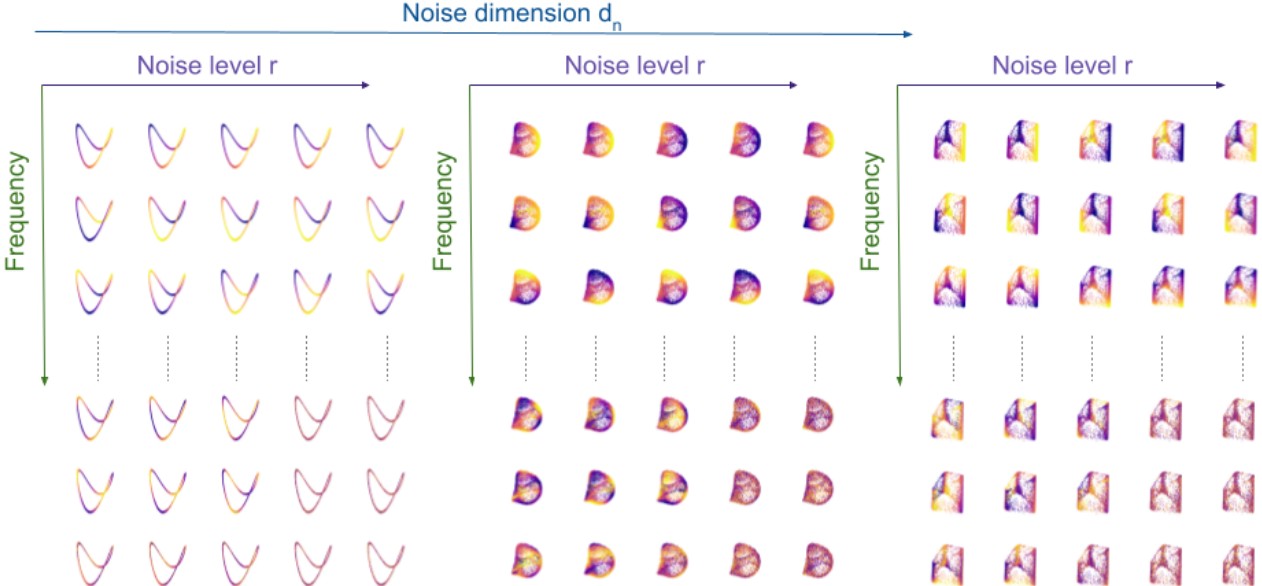

Figure 12: Stability w.r.t to $r$ of the low-frequency eigenvectors for the three real MDS datasets: Toluene(Left), Malonaldehyde(Middle), and Ethanol(Right). We color the coordinate spaces of the manifolds(for the MDS data these will be the $\phi_{1,2,3}(\Delta_{\mathcal{M}})$ by low-frequency(top) and high-frequency(bottom) $\phi_i(\Delta_{T_r(\mathcal{M})})$. Notice the stability of the former as $r$ increases and the corruption of the latter once the noise threshold is passed. In all plots $d_n = 50$ and, as opposed to the synthetic data, the noise is added in all ambient directions.

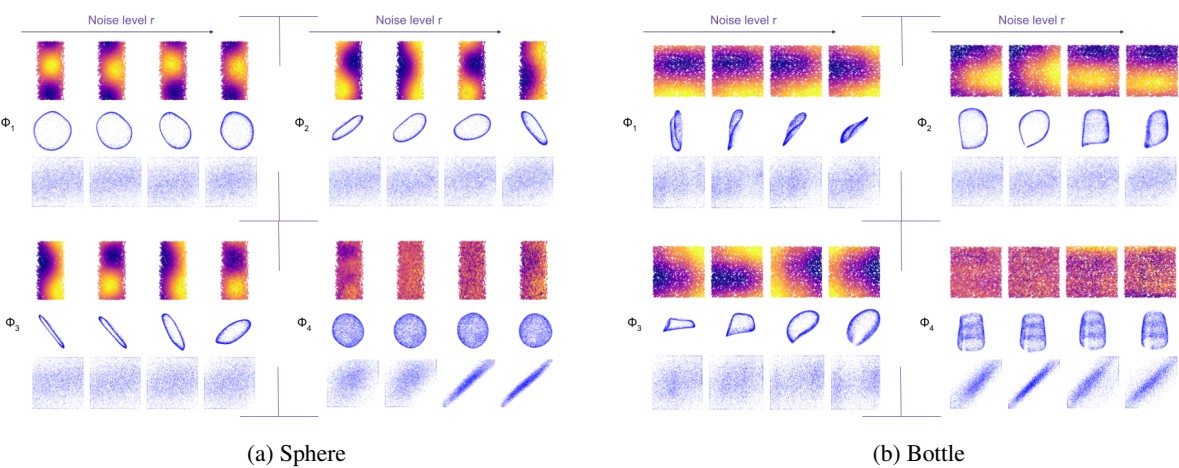

(a) Sphere  (b) Bottle

Figure 13: Results for VAEs trained on $S^2$ and Bottle for $d_n = 16$ and $r$ increasing from left to right. Each panel corresponds to one $\phi_i$ and displays from top to bottom: the coordinate space of the manifold colored by $\phi_i$, the reconstruction of the data without $\phi_i$, and the prediction of each $\phi_i$ from normal space/noise coordinates by a polynomial regression model. $\phi_{1,2,3}$ are stable w.r.t. to noise, indispensable for reconstruction, and cannot be predicted from noise. Conversely, $\phi_4$ is not correlated with the coordinate space of the manifold, not important for reconstruction, and easy to predict from noise coordinates.

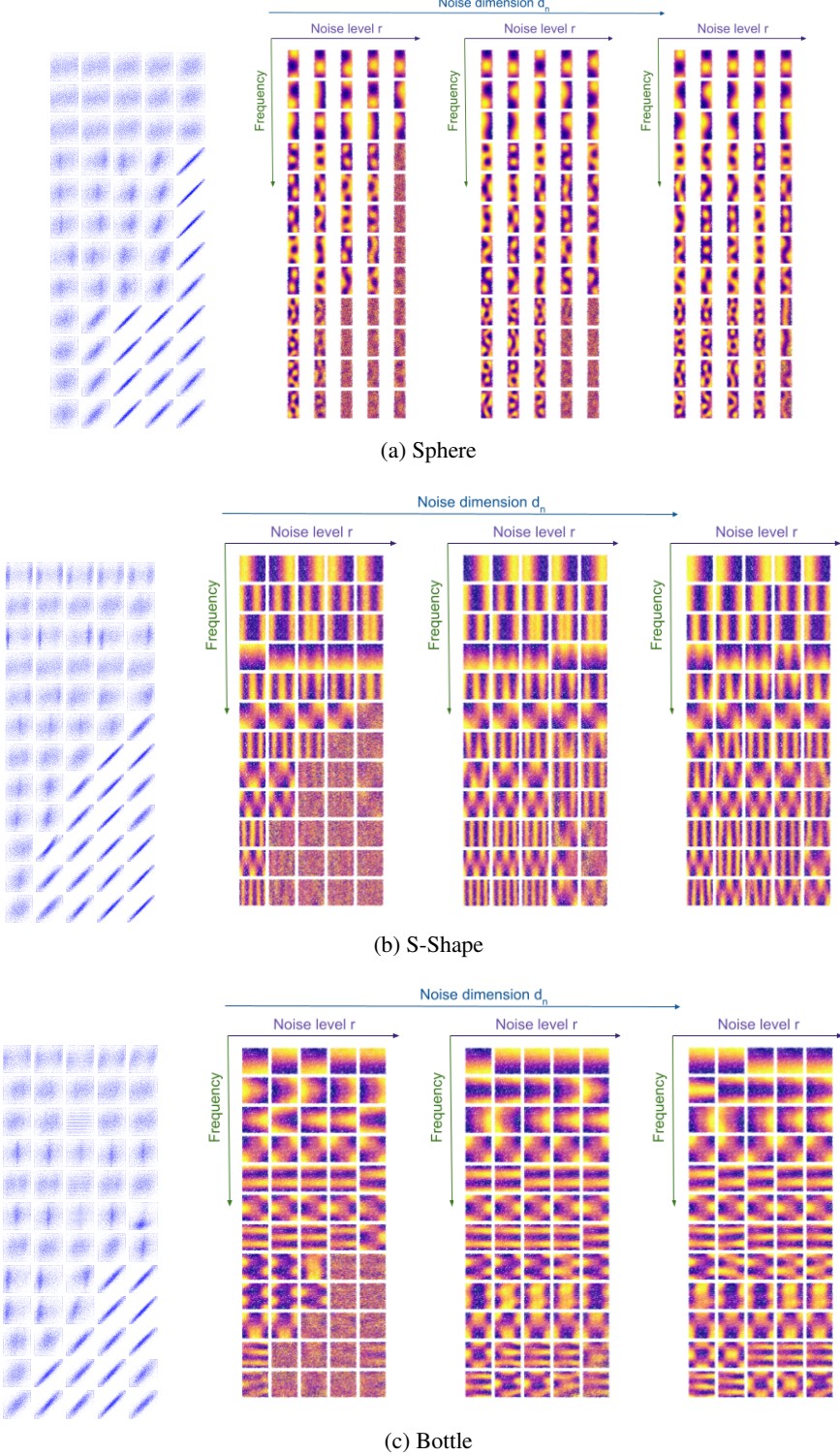

(a) Sphere

(b) S-Shape

(c) Bottle

Figure 14: Stability w.r.t to $r$ and $d_n$ of the low-frequency embeddings and the threshold effect exhibited by the high-frequency ones obtained using LTSA for three synthetic manifolds: Sphere(a), S-Shape(b), and Bottle(c). We color the coordinate spaces of the manifolds by $\phi_i$. Notice the similarity with Figure 11. As before, $r$ increases from left to right, while on each column we have $d_n \in \{16, 24, 32\}$. On the left-hand side we use a polynomial regression model to predict the $\phi_i$'s from the known normal space/noise coordinates for all $r$'s and $d_n = 16$. When we are able to do so(the diagonal lines tell us when that is the case) we can conclude that the corresponding $\phi_i$ represents pure noise.

