# OpenReview forum: "The Noisy Laplacian: a Threshold Phenomenon for Non-Linear Dimension Reduction"
_ICML.cc/2025/Conference — ICML 2025 poster_

### Official Review · Reviewer_Jzfg · 2025-03-09

**Overall Recommendation:** 3

**Summary:**

The authors focus on manifold learning and the effect of noise on the Laplacian operator in that context.
The paper is mainly a theretical study with some experiments to back up the claims.
The scope of the experiments is rather limited as they must fulfill strong assumptions (manifold data, uniform noise).

**Claims And Evidence:**

Manifold recovery in spite of noise.
First use of the Sasaki metric.
The main claims are reaonable and apparently proved in a sound way.

**Essential References Not Discussed:**

Good.

**Experimental Designs Or Analyses:**

Well illustrated experiments of manifold learning back up the claims.
Mainly synthetic data owing to the strong assumptions of the theory (manifold data only, constant/uniform noise level).
Some (very specific) real data is used in additional experiments.

**Methods And Evaluation Criteria:**

Looks sound (mainly theoretical proofs).

**Other Comments Or Suggestions:**

/

**Other Strengths And Weaknesses:**

The paper is well written and relatively easy to read for a theoretical one.
As most theoretical papers, the contributions are mainly proofs that have little impact on the practice.
The domain of interest (manifold learning with spectral embedding) is a niche (compared to DR with NE, e.g.).
These minor shortcomings do not decrease the value of the contribution for a slected audience, most probably.

**Questions For Authors:**

Can you refine your assesment LTSA? Same behavior but really no link? LLE not using explicitly a Laplacian but does it implicitly, as shown later. Possible to come to a similar result?

**Relation To Broader Scientific Literature:**

Good.

**Theoretical Claims:**

No.
The claims are mainly theoretical/mathematical and they largely exceed my paygrade.
As said above, the claims sound reasonable and there is no real surprise in the results (rcovery wrt noise level).

---

> ### Author Rebuttal · Authors · 2025-04-01
>
> We thank the reviewers for the constructive reviews! Geometric Data Analysis (GDA) is a small area, and your attention to it is appreciated.
>
> Here we respond to the main points raised by all reviewers.
>
> Is the result suprising and new in its particular area?
>
> Our sharp threshold result is actually surprising, because in the noiseless case, when samples are _on_ a manifold, the degradation in the eigenvectors is gradual, with no threshold. We show that if noise is added to the samples, the degradation is catastrophic at a predictable threshold. This was not expected from existing theory.
>
> Moreover, this threshold depends ONLY ON THE NOISE, not on other (unknown) properties of the manifold. This is also surprising, since so many other DiffMap properties depend on injectivity radius, reach, volume, etc.
>
> Third, it also suggests that *anisotropic noise* may paradoxically make the estimation of the eigenfunctions easier, due to widened spectral gaps.
>
> Finally, it adds to other indirect evidence that estimating a manifold by the eigenfunctions of the Laplacian, even though beautiful theoretically, may not be robust in practice. The authors plan to focus on LTSA in future work. [Note that embeddingless methods to estimate a manifold exist, their behavior is well studied, and _different_ from what we discover in this submission.]
>
> Practical implications:
> 1) Our result is an __impossibility results__. We show that estimating the Laplacian e-vectors has informational limitations, __even under strong assumptions__.  An impossibility result directly impacts further analysis under weaker assumptions.
>
> 2) If noise can be estimated, then from our analysis, the threshold will be known, and we will know which e-vectors belong to the manifold.
>
> 3) The experiments on VAE suggest that the threshold phenomenon we discovered is more general, and this begs to be known.
>
> 4)  UMAP the most popular of Neighbor Embedding methods uses the eigenvectors of the Laplacian to seed their embedding. Thus our result may be relevant to the UMAP users. More precisely, depending on how one uses UMAP, and how one avoids other unrelated artefacts of this heuristic, the threshold may become relevant or not.
>
> We will include comments 1) and 2) in the final version of the paper to clarify the implications of our study.
>
> We are aware this work is of a somewhat "niche field", and we thank the reviewer who expressed this fact. Indeed, Neighbor Embeddings are vastly more common. This reality will color the decision on this paper, whether it's explicitly stated or not.
>
> We understand that ICML may make the strategic decision to favor papers based on their area. When such a decision is made, it is not just affecting the authors, it is also implicitly telling the people who might read this paper that ICML the flaghips ML conference has no place for GDA (for example).
>
> More to the point, the theoretical properties we uncover are broadly relevant. As we have already discussed, our results have immediate importance for popular algorithms such as UMAP, and empirically we see similar threshold phenomena may affect other non-Laplacian embeddings. Thus our work is not only a result regarding one specific spectral analysis of a dataset, but a new informative perspective connecting denoising and dimension reduction.
>
> We now focus on reviewer specific points.
>
> We believe our work has many practical implications, and we will dwell on its implications for DM embeddings in particular. The sharp threshold phenomenon indicated by our results gives clear guidance to practitioners about the limitations of Laplacian embeddings. In practice, one may argue that this is already well understood, as low dimensional embeddings are typically preferred. The benefit of our results is that they provide new rationale for why this is essential, and moreso, how one may select a cut-off. In future research, techniques to analyze the structure of the noise may be developed to estimate exact cut-offs. Further, our theory suggests that ``uninformative" eigenfunctions catastrophically deviate from the underlying manifold structure, thus standard techniques to promote parsimony are effective at removing this uninformative information.
>
> Still dwelling on this point, not all papers that consider Laplacian embeddings advocate for low-dimensional bases. For example, in [Green, Tibshirani, 2021], infinite dimensional bases are employed for non-parametric regression. Our work provides an interesting complementary perspective on this. For signals that have little dependence on the noise, our results indicate that there is very little benefit to exceeding the "noise threshold". This opens up many questions about how this can be leveraged to improve learning rates.
>
> It is an excellent point raised about more refined results being possible for LTSA, as a connection to the Laplacian exists as seen in works such as [Ting, Jordan, 2018]. We plan to pursue this in future research.

---

### Official Review · Reviewer_BJ7A · 2025-03-14

**Overall Recommendation:** 3

**Summary:**

This work brings of the interesting Sasaki metric into manifold learning and Laplacian Enginmap, and prosed theoretical analysis results to support low frequency eigen recovery under noise constrains defined in abstract.

Strengths: the high-level idea of introducing Sasaki metric, with both "tangent" and "normal" concept, or as "horizontally" and "vertically", and with informative recovery threshold, which is linked with noise amplitude.

Experimental results on both synthetic data and real data, are presented to support Theorem 3.1, and include results from LTSA (ref. 3) and VAE (ref. 2), which are not Laplacian decomposition-based methods.

References:

1. R. R. Coifman and S. Lafon, “Diffusion maps”, Applied and Computational Harmonic Analysis, 2006
2. D. P. Kingma and M. Welling, “Auto-encoding variational bayes,” 2022.
3. Z. Zhang and H. Zha, “Principal manifolds and nonlinear dimensionality reduction via tangent space alignment,” SIAM, 2004.

**Claims And Evidence:**

Yes

**Essential References Not Discussed:**

The following paper is one of the foundation works in the area of manifold learning, in particular for Laplacian methods, feel should be cited.

M. Belkin and P. Niyogi. Laplacian eigenmaps for dimensionality reduction and data representation. Neural Computation, 15:1373–1396, June 2003

**Experimental Designs Or Analyses:**

Laplacian and non-Laplacian based methods on both synthetic and real data are included in section 4 & 5 & appendix, all look reasonable.

**Methods And Evaluation Criteria:**

Yes

**Other Comments Or Suggestions:**

Questions: not sure if I missed this, the noise scale "r" is defined in the abstract and mentioned section 2 too, though, it seems not see the clear definition or guess mostly as section 2.1 (the part refer to ref. 6, 19) as is.

**Other Strengths And Weaknesses:**

As seems related, manifold intrinsic dimensionality estimation, not explicit discussed in this work right, and feel will be interesting to see discussions in this line.

For real data with noise, often even the rough intrinsic dimensionality is challenging, and consider low frequency eigen recovery with theoretical analysis support, then guess perhaps under certain conditions dimensionality estimation can be more reliable.

**Questions For Authors:**

NA

**Relation To Broader Scientific Literature:**

NA

**Theoretical Claims:**

I looked Theorem 3.1, Theorem 3.3, and appendix section B and C, though not very carefully for examination.

---

> ### Author Rebuttal · Authors · 2025-04-01
>
> We thank the reviewers for the constructive reviews! Geometric Data Analysis (GDA) is a small area, and your attention to it is appreciated.
>
> Here we respond to the main points raised by all reviewers.
>
> Is the result suprising and new in its particular area?
>
> Our sharp threshold result is actually surprising, because in the noiseless case, when samples are _on_ a manifold, the degradation in the eigenvectors is gradual, with no threshold. We show that if noise is added to the samples, the degradation is catastrophic at a predictable threshold. This was not expected from existing theory.
>
> Moreover, this threshold depends ONLY ON THE NOISE, not on other (unknown) properties of the manifold. This is also surprising, since so many other DiffMap properties depend on injectivity radius, reach, volume, etc.
>
> Third, it also suggests that *anisotropic noise* may paradoxically make the estimation of the eigenfunctions easier, due to widened spectral gaps.
>
> Finally, it adds to other indirect evidence that estimating a manifold by the eigenfunctions of the Laplacian, even though beautiful theoretically, may not be robust in practice. The authors plan to focus on LTSA in future work. [Note that embeddingless methods to estimate a manifold exist, their behavior is well studied, and _different_ from what we discover in this submission.]
>
> Practical implications:
> 1) Our result is an __impossibility results__. We show that estimating the Laplacian e-vectors has informational limitations, __even under strong assumptions__.  An impossibility result directly impacts further analysis under weaker assumptions.
>
> 2) If noise can be estimated, then from our analysis, the threshold will be known, and we will know which e-vectors belong to the manifold.
>
> 3) The experiments on VAE suggest that the threshold phenomenon we discovered is more general, and this begs to be known.
>
> 4)  UMAP the most popular of Neighbor Embedding methods uses the eigenvectors of the Laplacian to seed their embedding. Thus our result may be relevant to the UMAP users. More precisely, depending on how one uses UMAP, and how one avoids other unrelated artefacts of this heuristic, the threshold may become relevant or not.
>
> We will include comments 1) and 2) in the final version of the paper to clarify the implications of our study.
>
> We are aware this work is of a somewhat "niche field", and we thank the reviewer who expressed this fact. Indeed, Neighbor Embeddings are vastly more common. This reality will color the decision on this paper, whether it's explicitly stated or not.
>
> We understand that ICML may make the strategic decision to favor papers based on their area. When such a decision is made, it is not just affecting the authors, it is also implicitly telling the people who might read this paper that ICML the flaghips ML conference has no place for GDA (for example).
>
> More to the point, the theoretical properties we uncover are broadly relevant. As we have already discussed, our results have immediate importance for popular algorithms such as UMAP, and empirically we see similar threshold phenomena may affect other non-Laplacian embeddings. Thus our work is not only a result regarding one specific spectral analysis of a dataset, but a new informative perspective connecting denoising and dimension reduction.
>
> We now focus on reviewer specific points.
>
> On dimension estimation. Thanks for bringing up this point. [Little, Maggioni, IEEE 2009] present a dimension estimation in noise algorithm. They show that when performing local PCA at a well chosen radius the singular values have the largest eigengap at $d$ equal to intrinsic dimension. In other words, the noise and geometric eigenvalues separate.
>
> Rather than directly estimating the dimension, it is also worth considering another perspective suggested by our paper. Dimension can be somewhat misleading. As seen in our analysis, high-dimensional manifolds can act as if they are low-dimensional if analyzed at an appropriate scale. Thus quantities such as covering numbers, or as popularly used in Laplacian estimation, median distance between datapoints, become more relevant in assessing the complexity of the data. If one was to assume that their samples were indeed generated from a tubular neighborhood, a simple estimator for intrinsic dimension could be the rate at which a random subsample covers the whole dataset when a relatively small number of subsamples is selected. As a dual perspective, for a generic dataset where one may not even adopt this strong tubular assumption, one may look at these same covering numbers and use them to assess an "intrinsic dimension" of the data.
>
> On the citation of Belkin and Niyogi, we agree that this is more than appropriate to reference in our paper, and we will gladly include it. Regarding $r$, as mentioned by the reviewer, this is defined in section 2, however we can more clearly indicate that it is the radius of the noise where appropriate.

---

### Official Review · Reviewer_j76L · 2025-03-16

**Overall Recommendation:** 3

**Summary:**

the authors provide a theoretical analysis that shows that Laplacian eigenfunctions capture the geometry of the underlying
manifold, without needing the noise amplitude or dimension to vary with the same size.  The main technique leverages the so called Sasaki metric in Riemannian geometry. They conduct experiments and also observe similar behavior in other non Laplacian based dimension reduction methods.

**Claims And Evidence:**

yes, the claims are generally supported. The mathematics is, to the best of my ability, sound. The experiments, while not exhaustive, do support the claims of the authors.

**Essential References Not Discussed:**

How does the current work related to prior work on L-infinity convergence?

SPECTRAL CONVERGENCE OF GRAPH LAPLACIAN AND HEAT KERNEL
RECONSTRUCTION IN L∞ FROM RANDOM SAMPLES by dunson, wu and wu

**Experimental Designs Or Analyses:**

Yes, I checked the soundess of the experimental design. They are fine.

**Methods And Evaluation Criteria:**

yes. The experiments demonstrated the cutoff phenomenon as predicted by the theory.

**Other Comments Or Suggestions:**

/

**Other Strengths And Weaknesses:**

The main weakness of this paper is motivation and readability.

Since most ML readers are likely not familiar with the Sasaki metric, I suggest the authors spend more time and text to introduce the metric and how it differs from traditional ones and provide more intuition.

**Questions For Authors:**

/

**Relation To Broader Scientific Literature:**

This is a theory paper that provides advancement on the theoretical analysis of popular algorithms that are broadly used in scientific applications.

**Theoretical Claims:**

I only checked the main statements of the theorems in the main text. Those theorems appear sound to me.

---

> ### Author Rebuttal · Authors · 2025-04-01
>
> We thank the reviewers for the constructive reviews! Geometric Data Analysis (GDA) is a small area, and your attention to it is appreciated.
>
> Here we respond to the main points raised by all reviewers.
>
> Is the result suprising and new in its particular area?
>
> Our sharp threshold result is actually surprising, because in the noiseless case, when samples are _on_ a manifold, the degradation in the eigenvectors is gradual, with no threshold. We show that if noise is added to the samples, the degradation is catastrophic at a predictable threshold. This was not expected from existing theory.
>
> Moreover, this threshold depends ONLY ON THE NOISE, not on other (unknown) properties of the manifold. This is also surprising, since so many other DiffMap properties depend on injectivity radius, reach, volume, etc.
>
> Third, it also suggests that *anisotropic noise* may paradoxically make the estimation of the eigenfunctions easier, due to widened spectral gaps.
>
> Finally, it adds to other indirect evidence that estimating a manifold by the eigenfunctions of the Laplacian, even though beautiful theoretically, may not be robust in practice. The authors plan to focus on LTSA in future work. [Note that embeddingless methods to estimate a manifold exist, their behavior is well studied, and _different_ from what we discover in this submission.]
>
> Practical implications:
> 1) Our result is an __impossibility results__. We show that estimating the Laplacian e-vectors has informational limitations, __even under strong assumptions__.  An impossibility result directly impacts further analysis under weaker assumptions.
>
> 2) If noise can be estimated, then from our analysis, the threshold will be known, and we will know which e-vectors belong to the manifold.
>
> 3) The experiments on VAE suggest that the threshold phenomenon we discovered is more general, and this begs to be known.
>
> 4)  UMAP the most popular of Neighbor Embedding methods uses the eigenvectors of the Laplacian to seed their embedding. Thus our result may be relevant to the UMAP users. More precisely, depending on how one uses UMAP, and how one avoids other unrelated artefacts of this heuristic, the threshold may become relevant or not.
>
> We will include comments 1) and 2) in the final version of the paper to clarify the implications of our study.
>
> We are aware this work is of a somewhat "niche field", and we thank the reviewer who expressed this fact. Indeed, Neighbor Embeddings are vastly more common. This reality will color the decision on this paper, whether it's explicitly stated or not.
>
> We understand that ICML may make the strategic decision to favor papers based on their area. When such a decision is made, it is not just affecting the authors, it is also implicitly telling the people who might read this paper that ICML the flaghips ML conference has no place for GDA (for example).
>
> More to the point, the theoretical properties we uncover are broadly relevant. As we have already discussed, our results have immediate importance for popular algorithms such as UMAP, and empirically we see similar threshold phenomena may affect other non-Laplacian embeddings. Thus our work is not only a result regarding one specific spectral analysis of a dataset, but a new informative perspective connecting denoising and dimension reduction.
>
> We now focus on reviewer specific points.
>
> The convergence result by Dunson, Wu, Wu has several overlaps with our work. That paper establishes guarantees for Laplacian estimation through empirical samples.  As we establish properties of the continuum limit of these estimators, these results can be combined to assess the quality by which noisy samples from a manifold can be used to recontstruct intrinsic Laplacian data, although not without caveats:
>
> 1. Our main results establish $L^2(\mu)$ approximation guarantees, differing from the $L^\infty$ result of DWW. One potential path to allign these is to extend our analysis to an appropriate RKHS topology, namely a high order sobolev space, from which a convergence result would then imply the desired sup-norm convergence.
>
> 2. The results of DWW are not technically applicable to our setting. They assume data to be drawn from a closed manifold, a condition tube manifolds violate. Showing that their result hold in our setting is non-trivial, although it can most likely be addressed using standard techniques.
>
> 3. The rate of convergence would also be of key interest. We expect this to be a multi-scale phenomenon similar to the spectral growth we analyze in our paper. We expect an initial estimation rate on the order of the intrinsic dimension of the data, as has been shown in other results which relate Laplacian approximation to covering and packing numbers.
>
> Lastly, we thank the reviewer for their constructive feedback on the readability of the paper. We can adjust our presentation to emphasize the relationship between Sasaki and induced metrics in the main text rather than the appendix.

---

### Decision · Program_Chairs · 2025-05-01

**Decision:**

Accept (poster)

**Comment:**

This paper analyzes a threshold phenomenon in nonlinear dimension reduction using Laplacian Eigenmaps applied to noisy high-dimensional data. The main contribution is a theoretical result showing a sharp transition in embedding quality as a function of the noise-to-signal ratio. The analysis is mathematically sound, drawing on tools from random matrix theory and perturbation analysis, and it offers insight into the limitations of spectral methods under noise.

Reviewers found the core result technically solid, though some noted that the scope is narrow and the practical relevance could be more clearly articulated. The empirical evaluation is minimal, serving primarily as a sanity check. The authors are encouraged to discuss potential generalizations to other methods or noise models. Overall, while focused, the work contributes a well-posed result to the theory of spectral embeddings.